# Validation metrics for ice edge position forecasts

Arne Melsom[1], Cyril Palerme[1], and Malte Müller[1]

[1]Norwegian Meteorological Institute

**Correspondence:** Arne Melsom (arne.melsom@met.no)

**Abstract.**

The ice edge is a simple quantity in the form of a line that can be derived from a spatially varying sea ice concentration field. Due to its long history and relevance for operations in the Arctic, the position of the ice edge should be an essential element in any system that is designed to monitor or provide forecasts for the physical state of the Arctic Ocean and adjacent ocean
regions.

Users of monitoring and forecast products for sea ice must be provided with complementary information of the expected accuracy of the data or model results. Such information is traditionally available as a set of metrics that provide an assessment of the information quality. In this study we provide a survey of metrics that are presently included in the product quality assessment of the CMEMS Arctic Marine Forecasting Center sea-ice edge position forecast. We show that when ice edge
results from different products are compared, mismatching results for polynya and local freezing at the coasts of continents and archipelagos have a large impact on the quality assessment. Such situations, which occur regularly in the products we examine, have not properly been acknowledged when a set of metrics for the quality of ice edge position results have been constructed.

We examine the quality of ice edge forecasts using a total of 15 metrics for the ice edge position. These metrics are analyzed
in synthetic examples, as well as in selected cases of actual forecasts, and finally for a full year of weekly forecast bulletins. Using necessity and simplicity of information as a guideline, we recommend using a set of four metrics that sheds light on the various aspects of product quality that we consider.

Moreover, any user is expected to be interested in a limited part of the geographical domain, so metrics derived as domain-wide integrated quantities may be of limited value. Consequently, we recommend that metrics are also made available for
an appropriate set of subdomains. Furthermore, we find that the metrics' decorrelation time scales are much longer than the present forecast range. Hence our final recommendation is to include depictions of gridded mismatching of ice edge positions using maps for the integrated ice edge error.

# 1 Introduction

The ice edge location is a primary source of information for safe navigation in ice infested waters. The retreating sea ice in the Arctic Ocean has given rise to increased naval traffic in the region. The navigation distance from Northern Europe to the Far East is about 40% shorter using the northern sea route when compared to the length of the southern route *via* the Suez Canal. Hence, commercial shipping is becoming viable from an economic perspective due to the changing physical conditions (Ho, 2010; Schøyen and Bråthen, 2011). Our motivation is to provide the increasing number of operators in the Arctic region with easily comprehensible and robust information about the quality of relevant forecasts.

Basic computations of ice edge displacement in operational sea ice forecasts relative to observational products have *e.g.* been performed by Posey et al. (2015) and Melsom et al. (2011). Results for the ice edge position from seasonal ensemble forecasts have been examined by Zampieri et al. (2018) and Palerme et al. (2019). Dukhovskoy et al. (2015) examined five metrics for ice edge displacement, and based on sensitivity tests for scale, rotation, translation, and noise, their recommendation is to apply the Modified Hausdorff Distance.

Model results for sea ice concentration are frequently examined by presenting differences from corresponding observations, or results from other models, as shaded contours on maps, see *e.g.* Johnson et al. (2007), Arzel et al. (2006). In these and other studies, results for sea ice are often quantified by simple statistics for integrated quantities, notably sea ice extent (Massonnet et al., 2012). Statistics for sea ice extent is one of the quantities that can be derived from contingency tables for sea ice concentration categories (Carrieres et al., 2017). A sophisticated approach to examinations of results for sea ice extent has been proposed by Goessling et al. (2016) who introduced the integrated ice edge error (IIEE) as an objective score for differences in the position of the ice edge. An extension relevant for ensemble predictions was recently published (Goessling and Jung, 2018). Using this extension, Palerme et al. (2019) find that ECMWF SEAS5 seasonal forecasts (Johnson et al., 2019) that are initialized between April and September are more skillful than climatology for forecast ranges of 6-12 weeks.

The fractions skill score (FSS) metric was developed for small scale features in forecast systems, originally applied to convective precipitation in weather forecasting (Roberts and Lean, 2008). One purpose of the FSS is to provide an objective analysis of how the forecast skill changes as a function of horizontal scales, which is potentially relevant for skill assessments of the ice edge position. The FSS was designed for features whose spatiotemporal evolution can't be forecasted exactly but rather in a statistical sense.

The present examination of validation metrics for the ice edge position has been performed with an aim of improving the information of product quality for users of products available from the Copernicus Marine Environment Monitoring Service (CMEMS). CMEMS is the marine component of the European Union's Earth Observation Programme. CMEMS has been set up to meet today's climate and marine challenges by providing the public with observational multiyear and near real-time products, and reanalyses and forecasts from ocean circulation models, sea ice models, wave models and biogeochemical models. The information is integrated into an open and free catalog of products that is available from http://marine.copernicus.eu/.

CMEMS is presently organized as 15 production centers, eight of which process observational data from satellite and *in situ* platforms, and the remaining seven centers run and process results from numerical models. These groups of centers are referred to as *thematic data assembly centers* (TACs) and *monitoring and forecast centers* (MFCs), respectively.

One of the TACs is dedicated to observations of sea ice, mainly based on data from satellite-born instruments. Furthermore, three of the MFCs' model systems have their ocean circulation model coupled to sea ice models. These are the centers responsible for forecasts and reanalyses in the Baltic Sea (BAL MFC), the Arctic Ocean (ARC MFC), and the global oceans (GLO MFC). Sea ice can also occur in the Black Sea, but the relevant forecast center (BS MFC) presently has no sea ice product.

Information about the product quality is available for all CMEMS model products, provided as statistics for a variety of metrics which are calculated by comparing results with observational products. Relevant data for sea ice concentration and the position of the ice edge is available from satellite-born instruments. In this study we assess the quality of forecasted ice edge positions using a large number of metrics. The sensitivity of the metrics due to differences in observational products is also considered.

The present examination is organized as follows. In Sect. 2 we introduce the metrics used in our analysis: ice edge displacement metrics in 2.1, IIEE and derived metrics in 2.2, and FSS metrics in 2.3. Next, an idealized situation which is constructed to shed light on situations which lead to large differences between model results and observations is explored in Sect. 3. This issue is investigated in the context of sea ice forecasts from CMEMS ARC MFC in Sect. 4, where results for two forecast bulletins with different error characteristics are presented. Then, results for a full year of sea ice forecasts are given in Sect. 5. These results are discussed in Sect. 6, and our examination concludes with a recommended best practice for validation of sea ice edge forecasts in 6.3.

## 2   Definition of metrics

We consider metrics for offsets in ice edge position between two gridded products, *e.g.* with one product derived from observations and with the other from simulation results from a numerical coupled sea ice-ocean circulation model. In this section, the two products are referred to as $O$ and $M$, respectively. Below we associate grid cell quantities by lower-case indices, and integral properties by upper-case indices. Analogously, we separate Euclidean grid cell distance values and integral distance metrics' values by denoting these as $d$ and $D$, respectively.

Note that in our approach, ice edges are associated with areas due to their composition of sets of grid cells, rather than curves. The definitions that lead to edge displacement metrics below do not directly apply to one-dimensional curves. Several displacement metrics between pairs of curves are given by Dukhovskoy et al. (2015).

### 2.1   Ice edge displacement metrics

In order to compute ice edge displacement metrics the first step is to find the grid cells which constitute the ice edge in the gridded observations as well as in the model product. Let $c$ be the sea ice concentration, and let $c_e$ be the sea ice concentration

value that defines the ice edge (usually set to 0.15). Then, we take the ice edge to be constituted by the grid cells $[i, j]$ that meet the condition

$$c[i,j] \geq c_e \quad \wedge \quad \min\left(c[i-1,j], c[i+1,j], c[i,j-1], c[i,j+1]\right) < c_e \tag{1}$$

where $\wedge$ is the logical AND operator. Let $E$ be the ice edge. Ice edges $E_O$ and $E_M$ then correspond to the set of grid cells $e_o$ and $e_m$ that are returned by this algorithm step when applied to products $O$ and $M$, respectively. We also introduce the coordinate position of grid cell $[i,j]$ as $[x,y]$, and let $N_O$ be the number of edge grid cells in product $O$, and $N_M$ be the number of cells in product $M$.

Next, for each edge grid cell in each product, we find the distance to the nearest edge grid cell in the other product. Consider first the distance from an ice edge grid cell $[i_m^1, j_m^1]$ in the model product at the coordinate position $[x_m^1, y_m^1]$. Then, the displacement of the observed ice edge from this grid cell becomes

$$d_m^1 = \min\left(\forall e_o \in E_O : \left[(x_o - x_m^1)^2 + (y_o - y_m^1)^2\right]^{1/2}\right) \tag{2}$$

where $\forall$ is the FOR ALL operator and $[x_o, y_o]$ is the coordinate position of an ice edge grid cell in the observed product.

A variant is to consider any land/ocean boundary grid cell as included in the observed sea ice edge. When adopting this variation we refer to the observational product as $\hat{E}_O$, constituted by grid cells $\hat{e}_o$. We note that $E_O \in \hat{E}_O$. The corresponding displacement becomes

$$\hat{d}_m^1 = \min\left(\forall \hat{e}_o \in \hat{E}_O : \left[(\hat{x}_o - x_m^1)^2 + (\hat{y}_o - y_m^1)^2\right]^{1/2}\right) \tag{3}$$

We compute the displacement $d_o^1$ of a model ice edge from an ice edge grid cell in the observational product analogously. This is also done for $\hat{d}_o^1$ after $E_m$ has been expanded to $\hat{E}_m$ by including all land/ocean boundary grid cells.

We can now define a set of symmetric ice edge position metrics expressed as functions of the edge displacements. Here, a symmetric metric is a parameter whose value is independent of whether it is the observations or the model product that is the base of the analysis. We introduce four such metrics here, based on results for $d_m$ and $d_o$.

1. The root-mean-squared ice edge displacement:

$$D_{RMS}^{IE} = \frac{1}{2}\left[\left(\frac{1}{N_O}\sum_{n=1}^{N_O}(d_o^n)^2\right)^{1/2} + \left(\frac{1}{N_M}\sum_{n=1}^{N_M}(d_m^n)^2\right)^{1/2}\right] \tag{4}$$

2. The average ice edge displacement:

$$D_{AVG}^{IE} = \frac{1}{2}\left[\frac{1}{N_O}\sum_{n=1}^{N_O}d_o^n + \frac{1}{N_M}\sum_{n=1}^{N_M}d_m^n\right] \tag{5}$$

3. The ice edge displacement bias, here defined as positive when the ice edge in the model product is on the open ocean side of the ice edge in the observational product:

$$\Delta^{IE} = \frac{1}{2}\left[\frac{1}{N_O}\sum_{n=1}^{N_O}\frac{c_m[i_o^n, j_o^n] - c_e}{\|c_m[i_o^n, j_o^n] - c_e\|}d_o^n + \frac{1}{N_M}\sum_{n=1}^{N_M}\frac{c_e - c_o[i_m^n, j_m^n]}{\|c_e - c_o[i_m^n, j_m^n]\|}d_m^n\right] \tag{6}$$

where $\|x\|$ is the absolute value of $x$, and $c_o, c_m$ are the sea ice concentrations in the observations and model, respectively. Also, $[i_o, j_o]$ and $[i_m, j_m]$ denotes ice edge grid cells in the observations and model, respectively. One may construct situations where a denominator in Eq. 6 becomes 0. In reality, such cases will be very rare, and most of the time this will occur when edge grid cells in the two products overlap, *i.e.*, $d^n = 0$. In these cases, we set the fraction to 0.

5  4. The extreme ice edge displacement, also known as the Hausdorff distance:

$$D_H^{IE} = \max\big(\max(d_o), \max(d_m)\big) \tag{7}$$

where $d_o, d_m$ are the full sets of gridded displacements as given by Eq. 3.

Finally, substituting displacements $d$ in Eq.s 4–7 by $\hat{d}$ as given by Eq. 3 gives rise to a set of supplementary metrics $\widehat{D_{RMS}^{IE}}$, $\widehat{D_{AVG}^{IE}}$, $\widehat{\Delta^{IE}}$, and $\widehat{D_H^{IE}}$. We note that *e.g.* $\widehat{D_{RMS}^{IE}} \leq D_{RMS}^{IE}$.

## 10  2.2  IIEE metrics

Recently, the integrated ice edge error (IIEE) has been suggested as an alternative approach to quantifying the offsets between two ice edges (Goessling et al., 2016). The IIEE is computed from the area between the ice edges in the two products. For a gridded product with a grid cell size $a$, set

$$
\begin{aligned}
a^+ &= \begin{cases} a & \text{for grid cells where } c_m > c_e \,\wedge\, c_o < c_e \\ 0 & \text{elsewhere} \end{cases} \\
a^- &= \begin{cases} a & \text{for grid cells where } c_o > c_e \,\wedge\, c_m < c_e \\ 0 & \text{elsewhere} \end{cases}
\end{aligned}
\tag{8}
$$

15  Then, the area where the ice edge position in the model product is on the open ocean side of the observed ice edge is

$$A^+ = \sum_A a^+ \tag{9}$$

whereas the complementary situation with the observed ice edge on the open ocean side of the model edge covers the area

$$A^- = \sum_A a^- \tag{10}$$

(an illustrated example is provided in Sect. 3). The ice edge is here the perimeter of the sea ice extent area. Thus, $A^+$ is the
20  area where the ice extent in the model results overshoots the ice extent in the observations, and *vice versa* for $A^-$.

Two area metrics can then be constructed, as given by Goessling et al. (2016).

1. The integral score:

$$A^{IIEE} = A^+ + A^- \tag{11}$$

2. The bias score:

$$\alpha^{IIEE} = A^+ - A^-$$ (12)

Note that Goessling et al. (2016) also introduced additional area metrics which are not considered here.

The IIEE metrics defined in Goessling et al. (2016) are all provided for areas of sea ice, while no displacement metrics are introduced. Here, IIEE-based displacement metrics are derived by dividing the IIEE areas by an IIEE characteristic length scale. Below, we introduce two definitions of such a length scale.

Summary statistics in the form of a contingency table provide versatile information for validation of sea ice concentration results (Carrieres et al., 2017). After categories have been defined by a set of ranges in sea ice concentration, table cells will give areas with category match-ups. Here it is essential to have the sea ice concentration value that defines the ice edge as a value that separates two categories. The sea ice extent for each product is then found as the sum of the relevant rows and columns, respectively. The differences in sea ice extent (quantities $A^+$ and $A^-$) emerge from adding the areas in cells that corresponds to categories on different sides of the ice edge in the two products.

### 2.2.1 Edge length based IIEE displacement metrics

In order to provide scores that have the same dimension as those produced by the ice edge displacement metrics in Sect. 2.1, we here introduce metrics that arise when dividing the area metrics given by Eq.s 11, 12 with the ice edge length. Presently, the ice edge is given as a set of grid cells that were identified from Eq. 1. For simplicity we consider the case where the resolution in both horizontal directions is constant and equal, and write the grid cell size as $s$.

Consider the schematic example provided in Fig. 1. When calculating the length of the ice edge, we must account for the presence of diagonal edge grid cells. This is performed by looping all edge grid cells $e$ and counting the number of $[i, j]$ edge grid cell neighbours (*i.e.*, among [i-1,j], [i+1,j], [i,j-1], [i,j+1]) in the same product. If there are two or more neighbours, the edge grid cell contributes with a length $l^e = s$ (edge grid cells $e_c, e_d$ in Fig. 1). If there are no such neighbours, the edge length is set to the length of the diagonal, *i.e.*, $l^e = \sqrt{2}s$ (edge grid cell $e_a$). If there is exactly one such edge neighbour, the contribution becomes $l^e = 0.5 \cdot (s + \sqrt{2}s)$ (edge grid cells $e_b, e_e$). Note that by this definition "open ended" edge grid cells (*e.g.* adjacent to land; $e_a, e_e$) will contribute with a diagonal representation towards the open end.

The ice edge length in the observational product becomes

$$L_O = \sum_{e \text{ in } E_O} l_o^e$$ (13)

and the corresponding length in the model product is given analogously.

Two length metrics can now be derived from the corresponding area metrics.

1. The IIEE average displacement:

$$D_{AVG}^{IIEE} = \frac{2}{L_O + L_M} A^{IIEE}$$ (14)

2. The IIEE bias:

$$\Delta^{IIEE} = \frac{2}{L_O + L_M} \, \alpha^{IIEE} \tag{15}$$

Note that if there are no overlapping ice edge grid cells in the two products and if no IIEE area is bounded by dry grid cells or an open boundary, the length scale used for derivation of the displacement metrics given by Eq.s 14 and 15 is half the circumference of the IIEE areas.

### 2.2.2 Separation based IIEE displacement metrics

An alternative to the application of the scaling length $(L_O + L_M)/2$ in Sect. 2.2.1 is introduced in Sect. S1 in the Supplementary Information document. The alternative expression for the scaling length is solely dependent on the geometry of the IIEE areas. We then derive a supplementary set of displacement metrics that is analogous to the $D^{IE}$ metrics defined by Eq.s 4-7.

The definitions of metrics in Sect. S1 take dry grid cells adjacent to IIEE areas into account, which the scaling length definition in Sect. 2.2.1 does not. Hence, we adopt here the hatted notation as introduced in Sect. 2.1. The resulting displacement metrics defined in Sect. S1 are thus denoted as $\widehat{D_{RMS}^{IIEE}}$, $\widehat{D_{AVG}^{IIEE}}$, $\widehat{D_{MAX}^{IIEE}}$, and $\widehat{\Delta^{IIEE}}$.

### 2.3 Fractions skill score

We next consider the fractions skill score (FSS), as introduced by Roberts and Lean (2008). This metric was defined with the purpose of providing information on the impact of differences on small scales that can appear in results from high resolution observations and models. The FSS is computed for binary results, such as gridded hits and misses due to a criterion, from a pair of products (usually observations and model results). Values for FSS provide information of how the two products compare as a function of resolution. Representation on different resolutions are computed by integration onto coarser (larger) grid cells, and the binary results on the original grid become hit fractions on coarser grids. The FSS reaches its maximum value of 1 at resolution(s) where the representation of the two products are identical, and has a minimum value of 0 when no grid cells have overlapping non-zero values.

In the present context, we define hits as grid cells which are part of the ice edge as defined by Eq. 1, in both products. The probability of a grid cell-by-grid cell match up of the edge positions is expected to be reduced when the resolution is enhanced.

The presentation of FSS in this section is largely based on the Roberts and Lean (2008) article, adapted to representation of lines of grid cells rather than areas. We provide a relevant schematic example as Fig. 2, and we use this to illustrate some of the quantities that are introduced below.

Recall from Sect. 2.1 that we identified the sets of $N_O$ and $N_M$ grid cells $e_o$ and $e_m$ that constitute the ice edges $E_O$ and $E_M$ in products $O$ and $M$, respectively. We construct a binary gridded representation of the ice edge in product $O$ as

$$\lambda_o[i,j] = \begin{cases} 1 & \forall e_o \in E_O \\ 0 & \text{elsewhere} \end{cases} \tag{16}$$

so that $\sum \lambda_o = N_O$. The corresponding binary representation of the edge in product $M$, $\lambda_m$, is defined analogously. Next, for product $O$ we introduce the coarse grid cell ice edge fraction for a neighbourhood with an extent of $n$ grid cells as

$$\lambda_o^n[i^n, j^n] = \frac{1}{n^2} \sum_{k=0}^{n-1} \sum_{l=0}^{n-1} \lambda_o\left[i^n + k - \frac{n-1}{2}, \ j^n + l - \frac{n-1}{2}\right] \tag{17}$$

where $n$ is an odd number. Again, we define $\lambda_m^n$ analogously, and we note that $\lambda_o = \lambda_o^1$. In the example in Fig. 2, a neighbour-
5 hood extent of 3 grid cells is indicated by the thick grid lines and for this case, we find

$$\lambda_O^{n=3} = \frac{1}{9} \begin{pmatrix} 2 & 1 \\ 0 & 2 \end{pmatrix} \qquad ; \qquad \lambda_M^{n=3} = \frac{1}{9} \begin{pmatrix} 3 & 1 \\ 0 & 3 \end{pmatrix} \tag{18}$$

The mean square edge fraction error for a neighbourhood extent of $n$ grid cells becomes

$$\text{MSE}^n = \frac{1}{N_x^n N_y^n} \sum_{i^n=1}^{N_x^n} \sum_{j^n=1}^{N_y^n} \left[\lambda_m^n[i^n, j^n] - \lambda_o^n[i^n, j^n]\right]^2 \tag{19}$$

where $N_x^n$, $N_y^n$ are the number of the neighbourhood extent $n$ grid cells in the x and y directions, respectively. Following
Roberts and Lean (2008) we introduce a reference MSE value as the largest possible with the present extent of the edge grid cells

$$\text{MSE}_{\text{ref}}^n = \frac{1}{N_x^n N_y^n} \min\left\{ \left[\sum_{i^n=1}^{N_x^n} \sum_{j^n=1}^{N_y^n} \lambda_o^n[i^n, j^n]^2 + \sum_{i^n=1}^{N_x^n} \sum_{j^n=1}^{N_y^n} \lambda_m^n[i^n, j^n]^2\right], \right.$$
$$\left. \left[\sum_{i^n=1}^{N_x^n} \sum_{j^n=1}^{N_y^n} \left(1 - \lambda_o^n[i^n, j^n]\right)^2 + \sum_{i^n=1}^{N_x^n} \sum_{j^n=1}^{N_y^n} \left(1 - \lambda_m^n[i^n, j^n]\right)^2\right]\right\} \tag{20}$$

This expression is a worst case arrangement of hits and misses that takes into account situations where hits outnumber misses. This is a modification of the corresponding definition in Roberts and Lean (2008) whose Eq. 7 allowed for situations with
15 $\text{MSE}_{\text{ref}}^n$ exceeding 1.

For the skill score with the original $6 \times 6$ grid in Fig. 2 we have $\text{MSE}^{n=1} = 6/6^2$ and $\text{MSE}_{\text{ref}}^{n=1} = 12/6^2$, while for the $n = 3$ neighbourhood displayed by the thick grid lines we have $\text{MSE}^{n=3} = 2/(2 \cdot 9)^2$ and $\text{MSE}_{\text{ref}}^{n=3} = 9/(2 \cdot 9)^2$.

Now, the resolution-dependent fractions skill score is introduced as

$$\text{FSS}^n = 1 - \frac{\text{MSE}^n}{\text{MSE}_{\text{ref}}^n} \tag{21}$$

which has a value of 1 for a perfect forecast for neighbourhood extent $n$ ($\lambda_m^n = \lambda_o^n \ \forall \ i^n, j^n \ \Rightarrow \ \text{MSE}^n = 0$) and a value of 0 when $\lambda_m^n \cdot \lambda_o^n = 0 \ \forall \ i^n, j^n$ ($\Rightarrow \ \text{MSE}^n = \text{MSE}_{\text{ref}}^n$). Note that invoking the modified definition of $\text{MSE}_{\text{ref}}^n$ in Eq. 20 makes the $\text{FSS}^n$ metric symmetric in the sense that reversing the definition of hits and misses does not affect the $\text{FSS}^n$ score.

For the sample case in Fig. 2 we then find that $\text{FSS}^{n=1} = 1/2$ and for the $n = 3$ neighbourhood displayed by the thick grid lines we have $\text{FSS}^{n=3} = 7/9 \approx 0.78$.

Moreover, we note from Eq.s 19-21 that the FSS score will not change if we introduce a set of additional grid cells where neither product has an ice edge, provided that non-events dominate events (*i.e.*, the first term in Eq. 20 is used, here: that the number of nodes without an ice edge is larger than the number of edge nodes). This observation has consequences for two different aspects in the present study.

First, when modeling the ocean, dry nodes are usually not considered to be part of the computational domain, and are assigned a special value in numerical results. When integrating over a neighbourhood $n > 1$ one option would be to discard the grid cells that are dry in the original representation. We will then be left with a result which has a non-constant neighbourhood size where $n^2$ if dry nodes are not present, and $< n^2$ for neighbourhoods where dry nodes are present. Here, we choose to avoid the problem of non-constant neighbourhood sizes by adopting $\lambda_o = \lambda_m = 0$ for dry grid cells.

Second, the grid for $n = 3$ indicated by thick lines in Fig. 2 is only one of nine possible configurations. Since the FSS results are not affected by additional grid cells where neither product has an ice edge, we can expand the original domain by adding a padding region of $n-1$ grid cells. In the case of $n = 3$ all configurations are attained by shifting the neighbourhood by 0, 1 and 2 original grid cells in both directions. The average FSS score from all of the configurations will be used henceforth in this article, since the alternative is a set of results that will depended on an arbitrary configuration subset choice.

As an expansion of the FSS metrics, Skok and Roberts (2018) introduced the FSS displacement, which we will refer to as $D^{FSS}$. An initial estimate for $D^{FSS}$ is derived by first determining for which neighbourhood size the FSS exceeds 0.5. The full algorithm for computing this displacement metric is given at the end of Skok and Roberts (2018), and is not repeated here. In most cases $D^{FSS}$ will become about half of the minimum metric neighbourhood size at which the FSS exceeds 0.5. The reliability of $D^{FSS}$ decreases when the frequencies are biased (Skok and Roberts, 2018). Here, this translates to differences

in the number of ice edge grid cells in observations and in the forecast. In the present study we implement a reduction of the product with the longest ice edge by randomly removing ice edge grid cells from this product. Thus, an unbiased version of the two grid cells is used when computing $D^{FSS}$. The random removal of grid cells is repeated a number of times, and the average value of the resulting displacements is taken to represent the $D^{FSS}$.

## 3   Ice edge metrics in two synthetic cases

In order to illustrate the various sea ice metrics and to examine how the results for these metrics compare, we have constructed a set of synthetic distributions of sea ice concentrations. The distributions will serve as representing observations and model results, respectively. The sea ice concentration distributions are introduced on a $200 \times 200$ grid, and they are displayed in Fig. 3.

We take the sea ice concentration field in Fig. 3a to represent a reference observation. One aspect of interest here is the effect on the validation scores when ice is introduced or removed locally in one product, but not in the other. In order to accentuate

such conditions, we supplement the reference observation with modified observation as displayed in panel b. A corresponding model result is given as shown in Fig. 3c.

We denote the comparison of the reference observation and model results as the *Reference case*, while the comparison of the modified observation and model results is referred to as the *Modified case*.

The ice edges (0.15 concentration isolines) as given by Eq. 1 are displayed as colored lines in Fig. 3. Edges from synthetic observations have been added in Fig. 3c. The main purpose of this article is to present metrics for the separation in such sets of lines.

Now consider the areas between the ice edges, from which the IIEE metrics are computed. The regions corresponding to the definitions in Eq.s 9 and 10 are shown in pink and red in Fig. 4.

The results for the various displacement metrics that were defined in Sect. 2 are given in Table 1. First, we note that in the *Reference case*, all $D^{IE}$ and $D^{IIEE}$ scores have similar values (with the expected exception of the maximum displacement score $D_H^{IE}$ which has a larger value than the other $D^{IE}$ scores by design). Also, $\Delta^{IE}$ and $\Delta^{IIEE}$ are of similar magnitudes in the *Reference case*.

For the modified case, we assume that the bottom boundary is adjacent to land. This is relevant for the hatted ice edge displacement metrics. From experience, we know that discrepancies where sea ice emerges or disappears at a distance from other ice covered regions arise from time to time in an operational sea ice forecasting service. An example will be presented in Sect. 4. We find that the value of the $D^{IE}$ ice edge displacement metrics given by Eq.s 4, 5 and 7 increase from the *Reference case* to the *Modified case* by a factor of about 2-5 even though a fairly modest area with additional sea ice has been introduced in the latter case. Since the additional discrepancy between the observations and model results has been introduced at a large distance, this change is according to our expectations.

Even though an additional discrepancy has been introduced in the *Modified case*, its shape and size is such that with the exception of bias metrics all IIEE displacement metrics increase by a very modest degree in these synthetic examples. In conclusion, we find that the deterioration according to scores for the *Modified case* is much larger for the $D^{IE}$ ice edge displacement metrics than for the IIEE metrics since the latter do not explicitly depend on the displacement between the pair of ice edges. Moreover, we note that if the ice edge displacement is defined by Eq. 3 the resulting $\widehat{D^{IE}}$ displacement increase only by a marginal fraction from the *Reference case* to the *Modified case*, due to the added ice area's proximity to land.

Finally, we note from Table 2 that the fractions skill score is only moderately reduced when additional observed sea ice is introduced locally in the *Modified case*, and the FSS displacement also increases modestly (Table 1, $D^{FSS}$). The changes in the IIEE area scores provide a quantification of the change in ice extent when substituting the *Reference case* with the *Modified case*.

A digression which is relevant here is that we have not included the Modified Hausdorff Distance, which was recommended by Dukhovskoy et al. (2015), in our analysis. In our formulation, this quantity is the maximum of the two terms in the bracket in Eq. 5, and will generally exhibit similar results to $D_{AVG}^{IE}$ but with larger magnitudes. While the sensitivity study in Dukhovskoy et al. (2015) is rich in detail, changes like contrasts between the *Reference case* and the *Modified case* are not considered. In their study of results from seasonal forecasts, Palerme et al. (2019) conclude that results for the Modified Hausdorff Distance are sensitive to differences with similar qualitative aspects as those discussed in this section. In Sect.s 4 and 5 below we will examine if differences which are qualitatively similar to the *Modified case* have an effect on the quality assessment of the ice edge position in the forecasts from CMEMS ARC MFC.

## 4 Ice edge metrics for two forecasts

We compare model results with observations which both are products that are distributed by CMEMS. The observational product is the Arctic Ocean Sea Ice Concentration Charts *Svalbard* which is a multi-sensor product that uses data from Synthetic Aperture Radar (SAR) instruments as its primary source of information (WMO, 2017). This product covers the northern Nordic
Seas, the Barents Sea and adjacent ocean regions. It is available on working days as mean values on a 1 km stereographic grid and will be referred to as the ice chart data hereafter.

Model results are taken from the Arctic Ocean Physics Analysis And Forecast product. Assimilation of sea ice concentration is implemented by use of microwave data, while no SAR data are assimilated. The model product will from here on be referred to as the ARC model product. In our investigation we will consider daily mean fields of sea ice concentration, which presently
are distributed on a 12.5 km stereographic grid. We restrict this study to the forecasts from the Thursday bulletins, which are available with a forecast range of ten days. The microwave data that are assimilated are available as the Ocean and Sea Ice Satellite Application Facility northern hemisphere product (Breivik et al., 2001), which is available from the CMEMS catalogue. The assimilation was performed three days prior to the Thursday bulletins. The main topic of this investigation is to provide an independent assessment of the quality of results for the ice edge, and not to assess the impact of assimilation. Thus,
we compare results with ice chart data rather than with the microwave data.

Prior to performing the analysis both products are regridded. The ice chart product is aggregated onto a 13 km grid, while the ARC model product is interpolated onto the same grid (the axes of the two CMEMS products, both available on polar stereographic grids, are rotated differently). The land-sea masks of the two regridded products are overlaid so that the geographical extent of the two regridded products is identical.

In order to explore how sea ice edge metrics from actual forecasts and observations are affected by changing conditions, we here examine two cases that illustrate contrasts of the type that was examined in Sect. 3. The two cases that are chosen are the day 5 ARC forecast products issued on 2017-03-30 and 2017-05-25. The quality of the forecasted ice edge positions will be assessed by comparing the model results with the ice edge position in the ice chart data on the respective forecast valid dates. The positions of the ice edges on these two dates according to model and observations are shown by displaying the IIEE fields
in Fig. 5a and b.

For the situation on 2017-05-29 (panel b) we notice that there are large discrepancies in the position of the ice edge in several locations: a polynya to the northwest of Greenland is open in the model, but not in the observations; there is a region along the coast in the Barents Sea where the model ice edge has retreated from the coast in the southern Kara Sea while the entire Kara Sea is frozen over in the ice chart; there remains some ice along the coast in the southeastern Barents Sea in the ice chart but
not in the model. These objects are indicated by labels in Fig. 5. Note also that polynyas have opened around Franz Josef Land (FJL), but since these are seen in both products this region doesn't affect the displacement metrics to the same degree as the other discrepancies that are mentioned here.

In contrast, the situation on 2017-04-03 (panel a) has notable offsets along the sea ice edge, but polynyas and mismatching results in coastal regions play a much smaller role than on 2017-05-29.

Results for the various displacement metrics are given in Table 3. As was seen in the results for the synthetic cases in Sect. 3, the scores that deviate substantially between the two forecasts are for the $D^{IE}$ ice edge displacement metrics and for $\Delta^{IE}$. The inflated values for the 2017-05-29 forecast when compared to the results for the 2017-04-03 forecast can largely be attributed to the ice edges associated with the IIEE features that are labeled in Fig. 5b. Furthermore, we note that the values for $\widehat{D^{IIEE}_{AVG}}$

and $\widehat{D^{IIEE}_{RMS}}$ are larger than those for the corresponding $\widehat{D^{IE}}$ metrics by a factor of 1.5-2. This contrast, which is much larger than in the synthetic case (Table 1), can be attributed to the fact that the individual IIEE features in the synthetic cases were few and regular. In the forecasts there is a large number of IIEE features with irregular shapes.

Furthermore, we find that the $\widehat{D^{IE}}$ metrics change only very modestly from 2017-04-03 to 2017-05-29 due to the proximity to the coast for the features that are labeled in Fig. 5b, in contrast to the results for $D^{IE}$. We also note that the definitions for the

displacement metrics that are derived from the IIEE lead to values for $\widehat{D^{IIEE}_{AVG}}$ that are about twice as large as the corresponding $D^{IIEE}_{AVG}$ values. Finally, we observe that for each of the two forecasts that are examined here, the relative difference between $D^{IIEE}_{AVG}$ and $\widehat{D^{IE}_{AVG}}$ is only about 10% or less. The relationships between the various displacement metrics are examined based on results from a full year of weekly forecast bulletins in the next section.

From the results for supplementary metrics in Table 4 we note that the FSS values are only slightly lower for the 2017-05-29

forecast than for the 2017-04-03 forecast, even though this forecast performs much poorer when diagnosed with the $D^{IE}$ ice edge displacement metrics.

## 5  Ice edge position metrics for 2017

The comparison of model results and observations in Sect. 4 has been performed for all weekly forecast bulletins from 2017. The results for mean displacement metrics and biases for the 5-day forecasts are displayed in Fig. 6. We note that there is a

seasonal variation in all metrics with large deviations during the months that lead up to the sea ice minimum in mid-September. We will refer to the period from the start of July to mid-September as the pre-minimum. A substantial part of the pre-minimum discrepancies is explained by the biases, which reveal that the sea ice extent is larger in the ice chart product than in the model product. The smaller extent in the model product gives rise to negative values in Fig. 6b. Annual average values for the various displacement metrics are given in rows *All 5-day forecasts* of Tables 3 and 4.

Furthermore, we note that the curves in Fig. 6 can be separated into two groups:

1. $D^{IE}_{AVG}$, $\widehat{D^{IIEE}_{AVG}}$ and $D^{FSS}$
2. $\widehat{D^{IE}_{AVG}}$ and $D^{IIEE}_{AVG}$

Group 1 metrics generally have larger values than group 2 metrics. This is expected since *e.g.* $\widehat{D^{IE}_{AVG}} \leq D^{IE}_{AVG}$ by definition, notably the different impact on these two metrics when the displacements occur in the vicinity of land or islands. Moreover,

we demonstrated in Sect. S1 that the definition of $\widehat{D^{IIEE}_{AVG}}$ in group 1 leads to values that are larger than the $D^{IIEE}_{AVG}$ metric in group 2.

Interestingly, we find that there is a contrast in the results between the two metrics groups during the pre-minimum: the deterioration exhibited in the evolution of group 1 metrics are larger than the corresponding deterioration for group 2 metrics in absolute terms. When we inspect the results from the two cases presented in Sect. 4, Table 3 reveals that the group 2 metrics have the lowest values in both cases. However, the separation into two distinct groups of metrics does not apply. We note that these two cases (indicated by vertical lines in Fig. 6) precede the July to mid-September pre-minimum during which the separation between the groups is most striking.

We have supplemented this analysis by a comparison between the microwave product that is assimilated by the model, and the ice charts. The deviations between these two observational products reveal similar peaks during the pre-minimum, *e.g.* with values for $D_{AVG}^{IE}$ and $\Delta^{IE}$ in ranges of about 60 - 120 km and -40 - -120 km, respectively (see Sect. S2 in the Supplementary Information document for details). Hence, the pre-minimum peaks that are seen in Fig. 6 can at least to some degree be attributed to assimilation of an observational product that deviates from the ice charts during the pre-minimum season. The correlation coefficients for the time series of $D_{AVG}^{IE}$ for the 5-day forecasts *vs.* ice charts (black line in Fig. 6a) and the time series of $D_{AVG}^{IE}$ for microwave data *vs.* ice charts is 0.89. The corresponding correlation coefficient for $\Delta^{IE}$ is 0.92.

Next, we have examined how the quality of the ice edge forecasts changes as a function of lead time. In order to limit the impact of the strong seasonal signal that is evident from Fig. 6, we have restricted this part of the analysis to the period from January to mid-May. The deterioration of the forecast quality that can be inferred from Fig. 7 is very weak. We also note that results for the two metrics in group 2 (blue and red curves in Fig. 6a) nearly overlap at all lead times, and are also lower in magnitude than the group 1 metrics at all lead times, as expected. The FSS scores for the same period are depicted as a function of resolution in Fig. 8, for model forecasts issued with a five day lead time, as well as for the microwave data. These results reveal that useful forecasts with a five day lead time are obtained at a scale of about 60x60 km, when the FSS reaches a value of 0.5 (which is a criterion recommended by Skok and Roberts (2016)). When comparing the microwave data with ice charts, the FSS is well above 0.5 for a neighbourhood extent $n = 3$, corresponding to useful data at a scale of approximately 40x40 km if ice chart data are taken as truth.

Finally, from the results in Table 4 we note that the model has a tendency to have a lower sea ice extent than the ice chart, as more than 70% of the IIEE areal misrepresentation is due to such conditions. This tendency is a confirmation of the negative bias values reported in Table 3.

## 6   Discussion

Our investigation of the results for the ice edge in the 2017 forecast bulletins in Sect. 5 revealed that the metrics $\widehat{D_{AVG}^{IE}}$ and $D_{AVG}^{IIEE}$ nearly overlap, and this is also the case for $\widehat{\Delta^{IE}}$ and $\Delta^{IIEE}$. These similarities can to some degree be understood from the following simplified cases: consider first a situation where one ice edge is shifted by a constant distance from the other, *i.e.*, they are parallel lines. Then, all of the average displacement metrics will be nearly identical, and this will also be the case for the displacement bias metrics. This is an idealized description for cases similar to the forecast for 2017-04-03 (Fig. 5a) where $D_{AVG}^{IE}$ is only moderately larger than $\widehat{D_{AVG}^{IE}}$ (Table 3).

Next, consider a situation where a part of one ice edge is shifted from the other, and the remaining part is due to discrepancies with coastal ice cover in one product, but not in the other. When the length of boundaries between IIEE areas and adjacent dry grid cells is much shorter than the ice edge length, the impact of disregarding coastal segments in Eq. 13 is small. Then, nearly identical displacement metrics values will again be the result for *e.g.* $\widehat{D_{AVG}^{IE}}$ and $D_{AVG}^{IIEE}$ by the same argument as above since the coastline will have taken on the role as an ice edge, or IIEE area limit. However, the value for $D_{AVG}^{IE}$ will inflate in this situation. These differences in displacement metrics will be further accentuated when such coastal discrepancies are separated geographically from the remaining ice edges as *e.g.* is seen with the labeled features in Fig. 5b, and $D_{AVG}^{IE} \gg \widehat{D_{AVG}^{IE}}$ (Table 3).

The main exception to the two types of situations described above occurs when polynyas form in the open ocean, away from the continental coasts as well as the Arctic islands. However, such cases rarely arise in the set of results that are investigated here.

Table 3 also includes results from a bootstrap analysis for the 2017 ice edge position metrics. The non-dimensional fractions that are listed are calculated by dividing the range spanned by the 5 and 95 percentile values by the mean value. Thus, smaller fractions indicate more robust results. We note that the fractions for the $D^{IE}$ metrics are larger than the fractions for the $\widehat{D^{IE}}$ metrics. The weakened robustness of the $D^{IE}$ metrics is due to the non-stationary behaviour of features that can give rise to inflated values for these metrics. Fraction values are not included for the bias metrics since bias averages can in principle be close to 0 with a combination of large positive and negative values. Hence, to complete the bootstrap analysis we here add that ranges spanned by the 5 and 95 percentile values for $\widehat{\Delta^{IE}}$ and $\widehat{\Delta^{IIEE}}$ are 9 km, while the corresponding ranges for $\Delta^{IE}$ and $\Delta^{IIEE}$ are 21 km.

## 6.1 Reducing the set of displacement metrics

The expected relationship between displacement metrics, conceptually described above, is confirmed by the results in Sect. 5. Hence, with the present configuration of validation domain and the results from model and observation, one in each of these two metrics pairs $\widehat{D_{AVG}^{IE}}, D_{AVG}^{IIEE}$ and $\widehat{\Delta^{IE}}, \Delta^{IIEE}$ can be disregarded. Of the two approaches, we find adopting $D_{AVG}^{IIEE}$ and $\Delta^{IIEE}$ to be the more intuitive and simpler choice (but admittedly this preference is somewhat subjective).

We can take this analysis one step forward, by systematically computing the correlation coefficients between all possible combinations of displacement metrics time series pairs. If we perform such an analysis for all 2017 forecasts and list the pairs whose correlation value is outside the range [-0.85, 0.85], 50 such pairs from a total of 105 pairs become listed. However, an influential seasonal cycle in the metrics, evident from the strong bias during the pre-minimum, has a sizable impact on the correlation results. If we instead restrict the analysis to the months prior to the pre-minimum, and retain the criterion that pairs with correlation outside [-0.85, 0.85] is of interest, we find that 13 of the proposed 15 metrics can be divided into four groups inside which metrics have large positive (> 0.85) or large negative (< -0.85) correlation coefficients. These groups are

1. All three $D^{IE}$ metrics
2. $D_{AVG}^{IIEE}, D^{FSS}, \widehat{D_{AVG}^{IE}}$
3. $\widehat{\Delta^{IE}}, \Delta^{IIEE}, \widehat{\Delta^{IIEE}}$

4. $\widehat{D_{RMS}^{IE}}, \widehat{D_{AVG}^{IIEE}}, \widehat{D_{RMS}^{IIEE}}, \widehat{D_{MAX}^{IIEE}}$

The two remaining displacement metrics are $\Delta^{IE}$ and $\widehat{D_H^{IE}}$.

Note also that the Hausdorff/maximum metrics are at times subject to large fluctuations depending on presence or absence of outliers. This was also noted in the investigation of skill metrics for sea ice model results by Dukhovskoy et al. (2015). Hence, a case can be made for disregarding the Hausdorff/maximum metrics.

## 6.2 Relative ice edge metrics

From the synthetic cases that were analyzed in Sect. 3, we note that the penalty for local freezing in one product but not in the other is much smaller for the IIEE-based displacement metric $D_{AVG}^{IIEE}$ than for the ice edge displacement metric $D_{AVG}^{IE}$. We therefore introduce two combined, relative metrics:

$$r_{AVG} = \frac{D_{AVG}^{IE}}{D_{AVG}^{IIEE}} \tag{22}$$

$$\widehat{r_{AVG}} = \frac{D_{AVG}^{IE}}{\widehat{D_{AVG}^{IE}}} \tag{23}$$

These derived metrics will *e.g.* increase in magnitude as local freezing are seen in the observational product and not in model results since the common numerator $D_{AVG}^{IE}$ will inflate. Then, if the model eventually becomes able to represent the local freezing, the metrics will decrease. For the synthetic cases we investigated in Sect. 3 we find $r_{AVG} = 1.03$ and $\widehat{r_{AVG}} = 1$ in the *Reference case*. In the *Modified case* we have $r_{AVG} = 1.82$ and $\widehat{r_{AVG}} = 1.90$. The corresponding set of ratios for the two forecasts that were examined in Sect. 4 are $r_{AVG} = 1.21$ and $\widehat{r_{AVG}} = 1.14$ on 2017-04-03, and $r_{AVG} = 2.89$ and $\widehat{r_{AVG}} = 3.17$ on 2017-05-29.

We started this discussion by noting that results for the two metrics which are the denominators in Eq. 22 and 23 nearly overlap. Hence, the curves in Fig. 9a also nearly overlap. However, this is not the case for the 5-day forecast for 2017-09-11, indicated by the rightmost vertical line in Fig. 9a. This outlier in the context of the metrics ratios can be explained by examination of the IIEE areas, for which the results in the Fram strait is shown in Fig. 9b. We can infer that there is a complex shape of a large part of the ice edge in the observational product (the red grid cells that have a blue neighbour) which is at some distance from the model ice edge. This inflates the edge-integrated metric $\widehat{D_{AVG}^{IE}}$ much more than the area-derived $D_{AVG}^{IIEE}$, and consequently $\widehat{r_{AVG}}$ (2.18) is significantly smaller than $r_{AVG}$ (2.94) in this case.

## 6.3 Recommendation

Our recommendations regarding a set of metrics to use for assessing the quality of ice edge forecasts are made from a preference of simplicity and necessity. By simplicity we have in mind metrics which are simple, not convoluted, in their implementation, and also have an intuitive interpretation. By necessity we have in mind a set of metrics for which each value provides useful information that is supplementary to the other values, and not overlapping.

From the analysis of validation results from a full calendar year that was presented in Sect. 5, and the subsequent discussion in 6.1 above, we recommend that validation results for ice edge displacement are provided for a set of three metrics:

1. $D_{AVG}^{IE}$

2. $D_{AVG}^{IIEE}$

3. $\Delta_{AVG}^{IIEE}$

Here, 1. and 2. give a high and a low bound for the expected displacement error for the position of the ice edge, respectively. The bias metric 3. provides information about whether the ice edge should be expected before or after a user reaches the forecasted position of the ice edge.

Moreover, while no new metrics are involved, we also encourage displaying results for

4. $r_{AVG}$

since time series for this quantity provides information on the robustness of the metrics results that can be easily presented as a line plot. In situations with large values of this fraction a user should be aware that the quality of the forecasted ice edge position is sensitive to how the displacement error is formulated. Note that of the two formulations in Eq. 22 and 23, our preference is the former since the episodic high impact of a complex ice edge makes interpretation of the latter less intuitive in the present context.

Another useful supplement when the pan-Arctic ice edge is considered is metrics statistics that are computed for sectors or sub-domains. IN CMEMS ARC MFC, we have adopted the Global Ocean Data Assimilation Experiment (GODAE; Bell et al., 2015; Hernandez et al., 2009) definitions of Arctic region when comparing forecasts to microwave observations. The GODAE Arctic regions are displayed in Fig. S3 in the Supplementary Information document. An alternative definition of Arctic sectors was adopted by Posey et al. (2015) in their quantification of the sea ice edge displacement.

Obviously, in a context of forecasting, validation results will always be available after the fact only. However, recent validation results are more often than not also relevant for a future period. We apply an auto-correlation crossing at $e^{-1}$ to define the decorrelation time scale. Then, we find that the decorrelation time scales of the metrics 1.-4. above are 6-7 weeks.

Frequently, users of forecast products are interested in the results for a small portion of the full domain. Hence, when possible validation results should be provided as easily accessible representations on maps. Hence, taking advantage of the long decorrelation time scale we recommend to supplement the above set of metrics with maps showing the distribution of IIEE areas (as *e.g.* Fig. 5).

This ends our recommendation for a basic set of ice edge displacement metrics. Nevertheless, more advanced users may also benefit from access to results for the FSS as a function of neighbourhood size: the FSS will also be highly relevant when performance changes due to increased resolution in model system upgrades are evaluated.

The above set of recommendations are based on an examination of results covering one year, for a specific forecast system and a specific observational product. While we believe that such an analysis is relevant for other sets of forecasts and observational products, each configuration should be checked separately, if resources are available. Issues like domain size (*e.g.*

pan-Arctic *vs.* regional) and resolution (representation of archipelagos and straits) can conceivably affect the characteristics of the forecast quality.

We end this study by noting that the travel time for commercial shipping between ports in Northwestern Europe and the Far East is about 20-30 days with speeds in the range 10-15 knots (5-7.5 m/s) (Schøyen and Bråthen, 2011). Adding a few days for advanced decision making of sea route, and subtracting some days for sailing time in ice free conditions at the end of the leg, forecast lead times of up to 20-30 day period is expected to be required in this context. Presently, CMEMS forecasts are available for lead times up to 10 days. We have shown that the deterioration in the forecast quality is moderate for these lead times (Fig. 7). Since maritime safety is one of the four core CMEMS areas of benefits, our final recommendation is to double the forecast lead time range of the CMEMS forecasting systems.

*Data availability.* All observational data that are used in this study is available from the CMEMS catalogue. The ice chart data and their documentation are available as product SEAICE_ARC_SEAICE_L4_NRT_OBSERVATIONS_011_002 from

http://marine.copernicus.eu/services-portfolio/access-to-products/?option=com_csw&view=details&product_id=SEAICE_ARC_SEAICE_L4_NRT_OBSERVATIONS_011_002,

and the microwave data and their documentation are available as product SEAICE_GLO_SEAICE_L4_NRT_OBSERVATIONS_011_001 from

http://marine.copernicus.eu/services-portfolio/access-to-products/?option=com_csw&view=details&product_id=SEAICE_GLO_SEAICE_L4_NRT_OBSERVATIONS_011_001.

The CMEMS ARC forecasts (product ARCTIC_ANALYSIS_FORECAST_PHYS_002_001_a) are also distributed from the CMEMS catalogue, but the forecasts are overwritten on a weekly basis by results from a delayed-mode ensemble simulation that is used for data assimilation purposes. The forecasts that are analyzed in this investigation is however publicly available from

http://thredds.met.no/thredds/myocean/ARC-MFC/myoceanv2-class1-arctic.html.

*Author contributions.* Melsom performed the analysis and wrote the article. Based on results from the analysis Palerme provided Fig.s 6 and 7, the remaining figures were provided by Melsom. Palerme and Müller contributed in discussions and provided comments and suggestions that significantly improved the presentation of the present study.

*Competing interests.* The authors declare that they have no conflict of interest.

*Acknowledgements.* We would like to express our gratitude to two anonymous referees whose comments and suggestions significantly improved our manuscript. We are also indebted to the model development team at the Nansen Environmental and Remote Sensing Center, and the provision of ice chart from the Norwegian Ice Service at the Norweian Meteorological Institute. This study has been performed on behalf of the the Copernicus Marine Environmental and Monitoring Service under Mercator Océan contract no. 2015/S 009-011301. Additional support was available from the Nansen Legacy project and the Salienseas project which are funded by the Norwegian Research Council under contracts no. 276730 and 276223, respectively. This is a contribution to the Year of Polar Prediction (YOPP), a flagship

activity of the Polar Prediction Project (PPP), initiated by the World Weather Research Programme (WWRP) of the World Meteorological Organization (WMO). Fig.s 1-5, 8-9, S1-3,5 were made using the NCAR Command Language (NCL, 2017).

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

| | Ice edge displacement metrics | | | | | | | |
|---|---|---|---|---|---|---|---|---|
| | $D_{AVG}^{IE}$ | $D_{RMS}^{IE}$ | $D_{H}^{IE}$ | $\widehat{D_{AVG}^{IE}}$ | $\widehat{D_{RMS}^{IE}}$ | $\widehat{D_{H}^{IE}}$ | $\Delta^{IE}$ | $\widehat{\Delta^{IE}}$ |
| *Reference case* | 9.1 | 10.6 | 20 | 9.1 | 10.6 | 20 | 0.24 | 0.24 |
| *Modified case* | 17.5 | 27.4 | 112 | 9.2 | 10.7 | 20 | -9.1 | -0.8 |

| | FSS | IIEE displacement metrics | | | | | | |
|---|---|---|---|---|---|---|---|---|
| | $D^{FSS}$ | $D_{AVG}^{IIEE}$ | | $\widehat{D_{AVG}^{IIEE}}$ | $\widehat{D_{RMS}^{IIEE}}$ | $\widehat{D_{MAX}^{IIEE}}$ | $\Delta^{IIEE}$ | $\widehat{\Delta^{IIEE}}$ |
| *Reference case* | 8.8 | 8.8 | | 10.4 | 10.5 | 10.6 | 0.17 | 0.21 |
| *Modified case* | 9.8 | 9.6 | | 11.0 | 11.1 | 13.4 | -1.7 | -2.3 |

**Table 1.** Results for the various displacement metrics defined in Sect. 2. Vertical lines are introduced to separate non-negative displacement metrics from signed bias metrics, and the FSS metric from IIEE metrics. The *Reference case* and the *Modified case* refer to the observational sea ice concentrations that are displayed in Fig. 3a and b, respectively. All values are given in non-dimensional grid units. Note that in the *Reference case*, all boundaries are considered open, and so the ice edge displacement metrics are unaffected when computing the hatted variables. Note also that in the *Modified case*, the bottom boundary was treated as adjacent to a closed (land) boundary.

|  | IIEE area metrics | | Fractions skill score | | |
|---|---|---|---|---|---|
|  | $A^{IIEE}$ | $\alpha^{IIEE}$ | $n=3$ | $n=7$ | $n=11$ |
| *Reference case* | 2002 | 38 | 0.14 | 0.26 | 0.37 |
| *Modified case* | 2470 | -430 | 0.12 | 0.24 | 0.34 |

**Table 2.** Supplementary metric scores. IIEE area scores are given in non-dimensional grid units. The fractions skill scores is computed by Eq. 21.

| | Ice edge displacement metrics | | | | | | | |
| --- | --- | --- | --- | --- | --- | --- | --- | --- |
| | $D_{AVG}^{IE}$ | $D_{RMS}^{IE}$ | $D_{H}^{IE}$ | $\widehat{D_{AVG}^{IE}}$ | $\widehat{D_{RMS}^{IE}}$ | $\widehat{D_{H}^{IE}}$ | $\Delta^{IE}$ | $\widehat{\Delta^{IE}}$ |
| *Forecast 4-3* | 35 | 47 | 150 | 31 | 43 | 150 | -14 | -15 |
| *Forecast 5-29* | 98 | 230 | 1560 | 31 | 39 | 130 | -87 | -23 |
| *All 5-day forecasts* | 69 | 116 | 720 | 37 | 48 | 175 | -55 | -27 |
| *Bootstrap fraction* | 0.25 | 0.27 | 0.28 | 0.17 | 0.15 | 0.15 | | |

| | FSS | | | IIEE displacement metrics | | | | |
| --- | --- | --- | --- | --- | --- | --- | --- | --- |
| | $D^{FSS}$ | $D_{AVG}^{IIEE}$ | | $\widehat{D_{AVG}^{IIEE}}$ | $\widehat{D_{RMS}^{IIEE}}$ | $\widehat{D_{MAX}^{IIEE}}$ | $\Delta^{IIEE}$ | $\widehat{\Delta^{IIEE}}$ |
| *Forecast 4-3* | 45 | 29 | | 61 | 69 | 100 | -14 | -40 |
| *Forecast 5-29* | 48 | 34 | | 57 | 61 | 91 | -27 | -48 |
| *All 5-day forecasts* | 61 | 39 | | 79 | 86 | 119 | -29 | -64 |
| *Bootstrap fraction* | 0.28 | 0.18 | | 0.18 | 0.18 | 0.15 | | |

**Table 3.** Results for the various sea ice edge displacement metrics. *Forecast 4-3* and *Forecast 5-29* results are metrics for the forecast for 2017-04-03 issued on 2017-03-30, and for the forecast for 2017-05-29 issued on 2017-05-25, respectively. *All 5-day forecasts* results are averages for all weekly 2017 forecast bulletins with a 5 day lead time. *Bootstrap fraction* is the difference between the 95 percentile and 5 percentile values from a bootstrap analysis of the *All 5-day forecasts* results, divided by the corresponding average value. All values are in km except the bootstrap fractions which are non-dimensional. See the text for details.

|  | IIEE area metrics | | Fractions skill score | | |
| --- | --- | --- | --- | --- | --- |
|  | $A^{IIEE}$ | $\alpha^{IIEE}$ | $n = 3$ | $n = 7$ | $n = 11$ |
| *Forecast 3-4* | 220 | -110 | 0.35 | 0.63 | 0.75 |
| *Forecast 5-29* | 210 | -167 | 0.30 | 0.54 | 0.68 |
| *All 5-day forecasts* | 260 | -186 | 0.30 | 0.49 | 0.59 |
| *Bootstrap fraction* | 0.21 | | 0.20 | 0.17 | 0.15 |

**Table 4.** Supplementary metric scores for the forecasts displayed in Fig. 5 and the corresponding 2017 average values. IIEE area scores are given in units of $1000\,\mathrm{km}^2$.

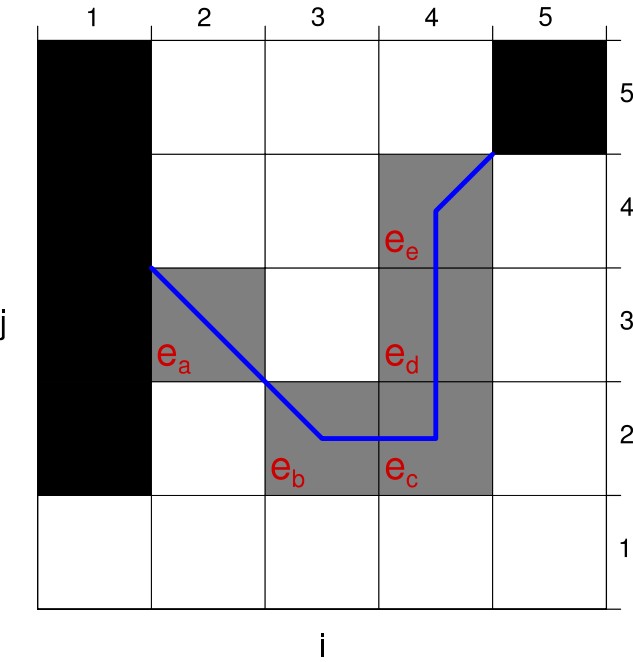

**Figure 1.** Schematic illustration for computation of ice edge length. The ice edge is displayed by the labeled cells that are filled grey. Black cells correspond to land. The algorithm we present here for calculation of the ice edge length yields a value that corresponds to the length of the blue line, see the text for details.

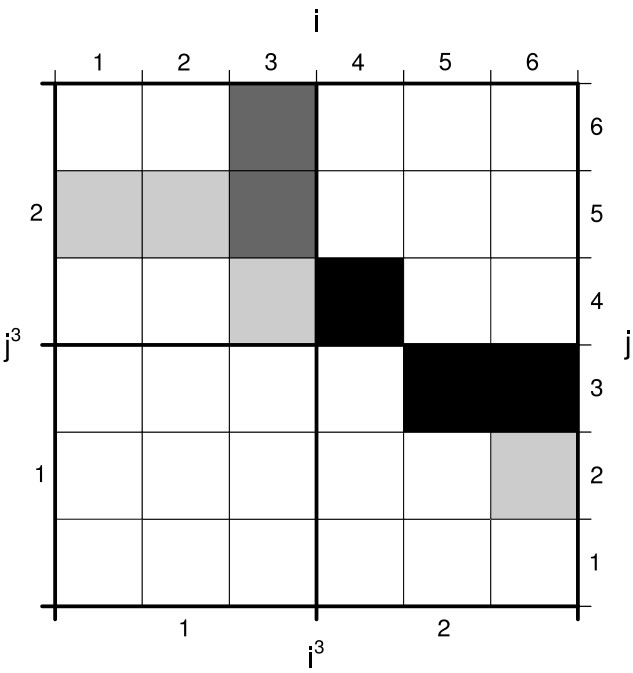

**Figure 2.** Schematic illustration for computation of fractions skill score for gridded contour lines. Gridded lines representing the ice edge of the model product and the observational product are shown as light gray boxes and dark gray boxes, respectively. Grid cells where the two lines overlap are black. The original grid is displayed by thin grid lines with $x$-axis indices at the top and $y$-axis indices to the right. Thick grid lines correspond to the grid of a neighbourhood with an extent of 3 grid cells ($n = 3$), with $x$- and $y$-axis indices at the bottom and to the left, respectively. See the text for details.

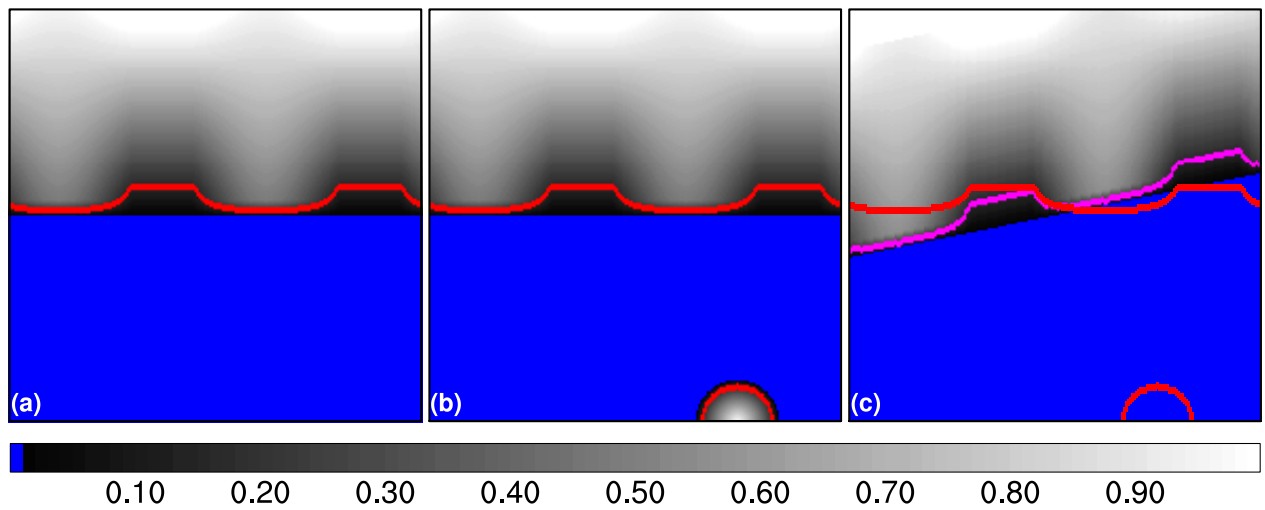

**Figure 3.** Sea ice concentrations representing (a) reference observations, (b) modified observations and (c) model results. The ice edges in the observational and model products are drawn as red and magenta lines, respectively. (These lines are drawn with three times their actual thickness in order to accentuate the edges graphically.) Note that the ice edge from the modified observations has been added in (c). Blue color represents ice free conditions, and the gray scale used for sea ice concentration is displayed by the label bar at the bottom.

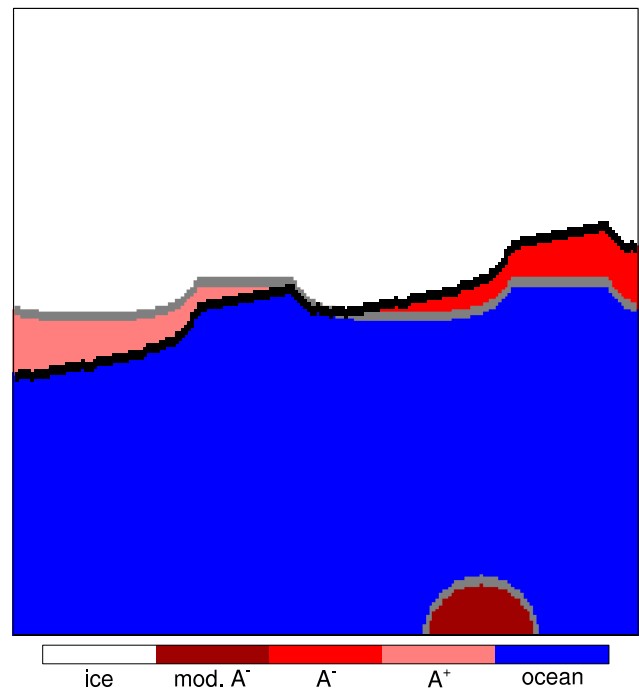

**Figure 4.** Depiction of areas used for computing the IIEE metrics. The pink region corresponds to the $A^+$ area given by Eq. 9, whereas the $A^-$ area given by Eq. 10 is in red. The additional $A^-$ area in the *Modified case* is in dark red. Ice edges are displayed as gray lines (observations) and black lines (model results). (These lines are drawn with three times their actual thickness in order to accentuate the edges graphically.) Regions where all products are on the open ocean side of the ice edges are blue, while regions which are inside the ice edges in all products are white.

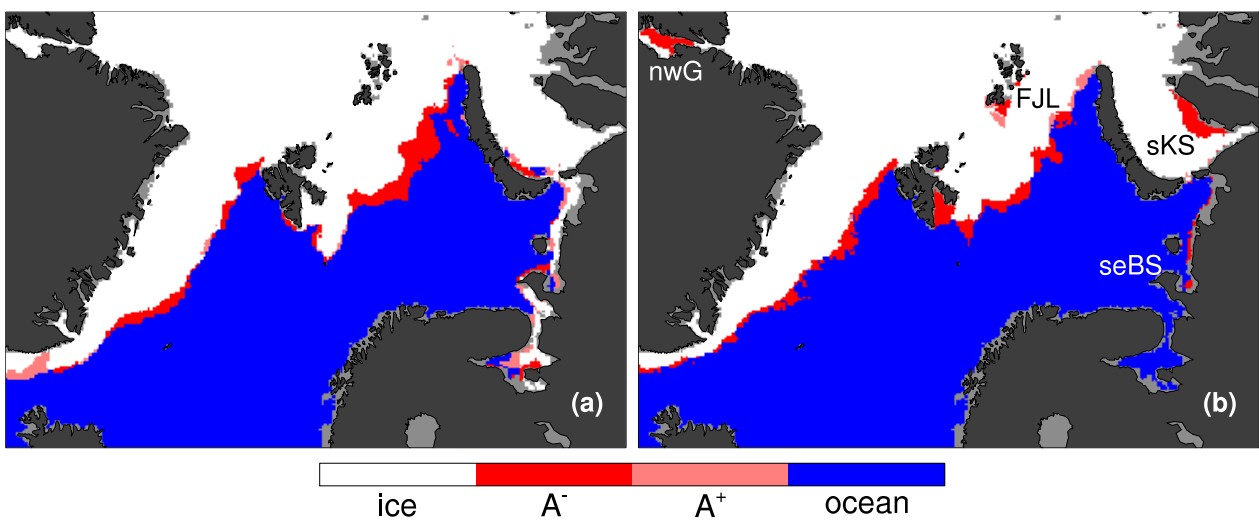

**Figure 5.** Map displaying the IIEE regions for two forecasts. Panels a and b display the results for the forecast for 2017-04-03 issued on 2017-03-30, and for the forecast for 2017-05-29 issued on 2017-05-25, respectively. Areas displayed in gray are not included in one or both products, and are excluded in the present analysis. The following regions with ice edge discrepancies are labeled in panel b: near Franz Josef Land (FJL), southern Kara Sea (sKS), northwest of Greenland (nwG), and southeastern Barents Sea (seBS). The displayed region is nearly the same as the region with ice chart data (a slight zooming was applied in order to highlight features of interest, so narrow bands of grid cells from the ice chart data to the right and to the bottom are not shown). The color codes for the various IIEE regions are the same as in Fig. 4.

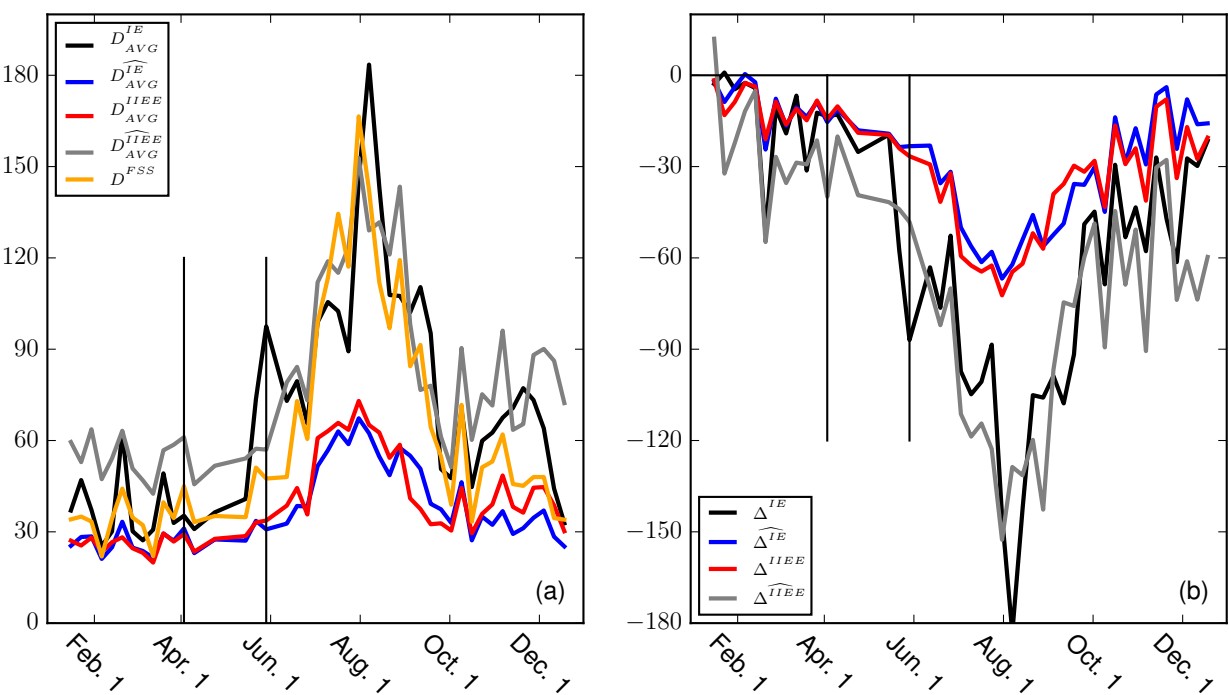

**Figure 6.** Time series for (a) mean displacement and (b) bias metrics as defined in Sect. 2. All results are for the 5-day forecasts. Vertical lines correspond to the two forecasts that were analyzed in Sect. 4. Values along the vertical axes are in units of km.

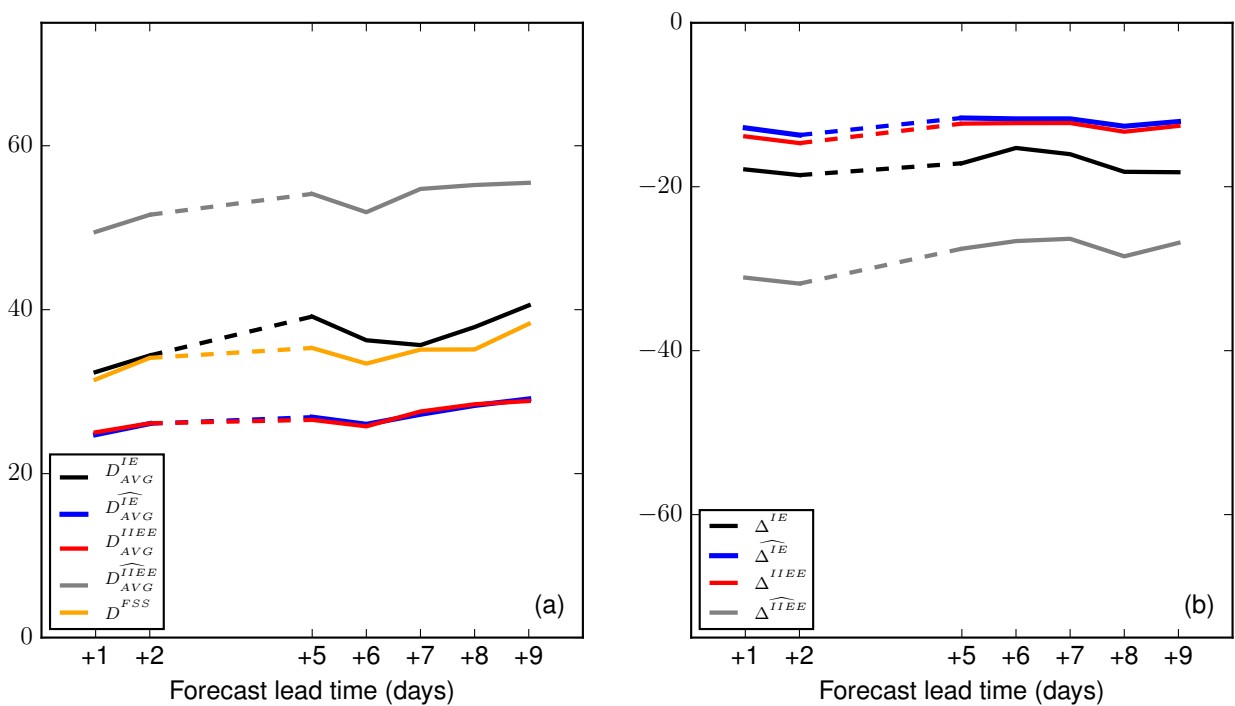

**Figure 7.** Metrics for (a) mean displacement and (b) bias, as functions of forecast lead time, in days. These results are based on forecast bulletins from January 2017 to mid-May 2017. Note that lines for $\widehat{D_{AVG}^{IE}}$ and $D_{AVG}^{IIEE}$ in (a) nearly overlap, as do lines for $\widehat{\Delta^{IE}}$ and $\Delta^{IIEE}$ in (b). Values along the vertical axes are in units of km. Ice charts are not produced on Saturdays and Sundays, which correspond to forecast lead times of +3 days and +4 days, respectively. Dashed lines are thus used to indicate the lack of analysis for these two days.

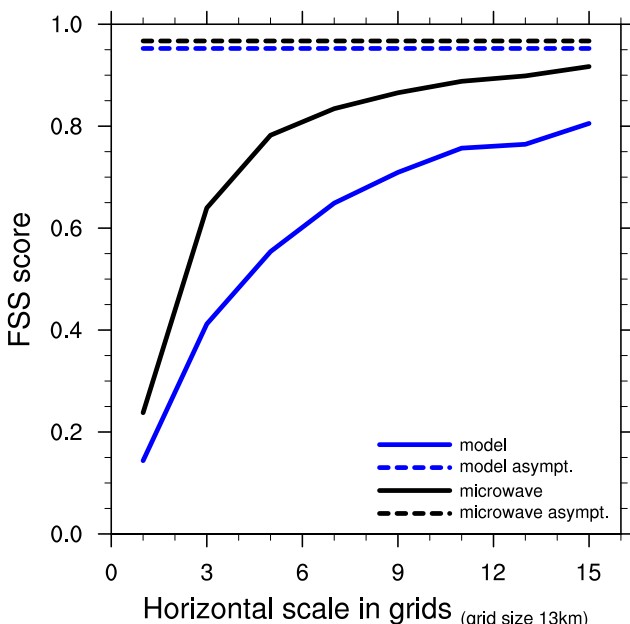

**Figure 8.** Fractions skill score as a function of resolution, for five-day lead time model forecasts *vs.* ice chart data (blue line) and microwave data *vs.* ice chart data (black line). Dashed lines show the asymptotic FSS values as defined by Roberts and Lean (2008) (their Eq. 8). These results are based on forecast bulletins and microwave data from January 2017 to mid-May 2017.

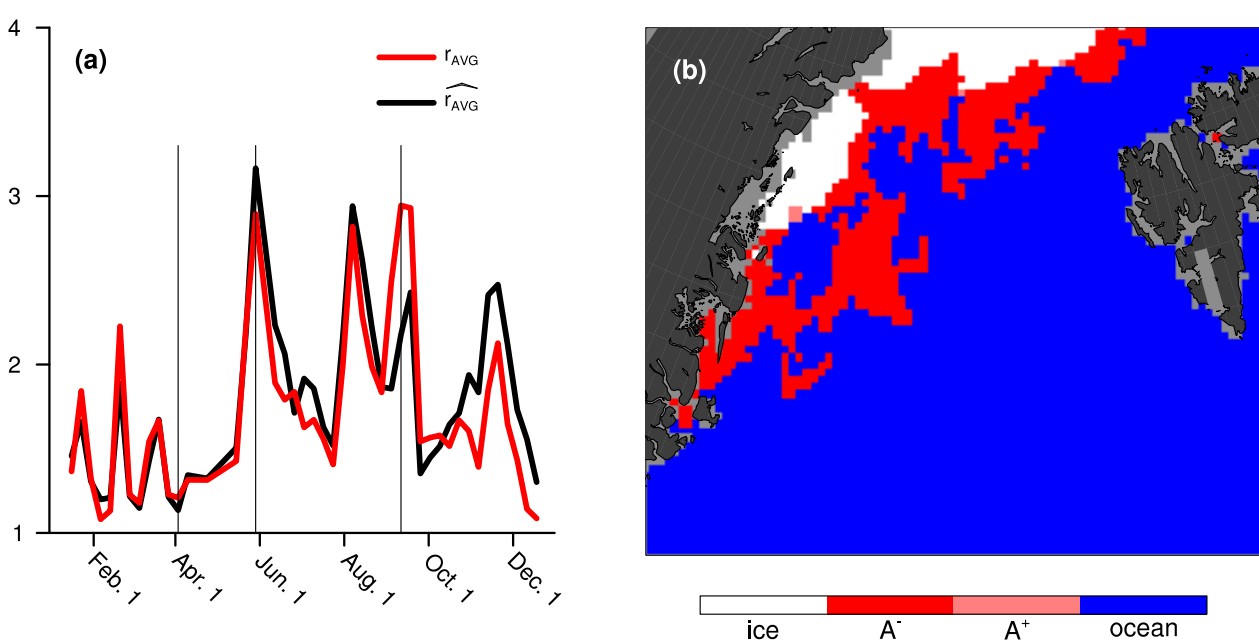

**Figure 9.** (a) Time series of two metrics ratios for forecasts with a lead time of 5 days. Vertical lines correspond to cases for which results are discussed in detail. The left and center vertical lines correspond to the two forecasts that were analyzed in Sect. 4, whereas the line to the right is for the situation displayed in the right panel. (b) Detail of IIEE in the Fram strait (the region between Greenland and the Svalbard archipelago) on 2017-09-11.