# Peer review of "Validation metrics for ice edge position forecasts"

_Ocean Science, 2018_

## Author Comment (AC1) · 4 Jan 2019

The x axis labels do not display correctly in Fig 4 and Fig 5. The corresponding author is sorry to have missed this in the original submission. The correct figures are attached here.
* * *
[Figure]

[Figure]

**Fig. 1.** Fig4

**Fig. 2.** Fig5

---

## Short Comment (SC1) · 24 Jan 2019

Formulae for Hausdorff distance and Modified Hausdorff distance have to be given. Eq. 7 is not complete, I couldn't find definitions of d0 and dm in the text. Please give an exact formula for HD that was used. I would strongly discourage using "extreme ice edge displacement" as a synonym for any of the Hausdorff distances. These are not the same. Extreme ice edge displacement is a separate metric based on the norm defined as max|dist(a,b)|, which has nothing to do with the HD or MHD. Please, do not confuse the readers. My understanding is that neither MHD nor HD has been really tested in this exercise, this should be stated in the document. In the text: " In our formulation, this quantity is the maximum of the two terms in the bracket in Eq. 5, and will generally exhibit similar results to DIE AV G but with larger magnitudes." From what I see, Eq.5 is not MHD.

---

## Author Comment (AC2) · 31 Jan 2019

**Authors' response to Short Comment 1 (D. Dukhovsky)**

We would like to thank Dr. Dukhovsky for taking the time to read our manuscript and provide his comments which we find very useful. We realize that some text should be rewritten, however, our understanding of the specific issues raised by the reviewer is that they are mostly misunderstandings of our manuscript.

We first acknowledge that terms like 'displacement' and 'distance' may not be intuitively regarded as mathematically precise terms. In our manuscript various definitions in Sect. 2 are given in order to define the terminology that we have adopted. When we have received all comments to our manuscript, we will revisit the topic of terminology.

We did not provide a definition of the term "ice edge displacement", this will be included in a revised manuscript version. Tentatively, our definition is likely to be:
"In the present manuscript the ice edge displacement is the shortest distance from an ice edge position in one product to the ice edge in another product."
Metrics are then defined as various functions of the gridded displacements, as given by Eq. (4)-(7).

If we denote an 'ice edge position in one product' by $a$ and the other product's full ice edge by $B$, the ice edge displacement as given above is then the distance denoted by $d(a, B)$ by Dukhovskoy et al. (their p. 5914). We reject the alternative definition that the ice edge displacement is the distance from an ice edge position in one product to any ice edge position in another product: we require that the ice edge displacement is 0 if the ice edges in two products overlap. (The comment we reply to refers to a $dist$ function which is not defined, hopefully the item has been covered by our response here.)

The comment that "d0 and dm" are not defined is incorrect ('d0' should be 'do'). We provide a definition of $d$ in Eq. (2), and the use of subscripting has been explicitly stated in the first paragraph of Sect. 2. Nevertheless, when we revise our manuscript we will add a sentence or two for Eq. (2) to make the link between this definition and $d_m, d_o$ even more clear.

Given the above definitions, our expression for the Haussdorf distance in Eq. (7) is identical to Eq. (10) in Dukovskoy et al. In order to remove any ambiguity, we will add a comment to the effect that $d_o, d_m$ are the full sets of gridded displacements as given by Eq. (2). Hence, results for the Haussdorf distance are provided and discussed in our manuscript.

The comment that the modified Haussdorf distance ($D_{MH}^{IE}$) has not been tested is correct, and we have not stated that we test this metric. Using our definitions in Sect. 2, the term defined as $D_{MHD}$ by Eq. (11) in Dukhovskoy et

al. becomes

$$D_{MH}^{IE} = \max\left[\frac{1}{N_O}\sum_{n=1}^{N_O} d_o^n, \frac{1}{N_M}\sum_{n=1}^{N_M} d_m^n\right]$$

using our terminology. Although we don't test this metric, we are aware of its use and discuss the close relation between $D_{MH}^{IE}$ as given here with $D_{AVG}^{IE}$ as given by our Eq. (5) (p. 8, l. 20-24).

**Reference**

Dukhovskoy, D. S., Ubnoske, J., Blanchard-Wrigglesworth, E. , Hiester, H. R., and Proshutinsky, A.:
Skill metrics for evaluation and comparison of sea ice models, J. Geophys. Res. Oceans, 120, 5910-5931, doi:10.1002/2015JC010989, 2015.

---

## Referee Comment (RC1) · Anonymous Referee #1 · 5 Feb 2019

\* Summary

Melsom et al. define a number of sea-ice forecast verification metrics for the Arctic ice-edge location and investigate pros and cons of these metrics, most of which are related to or identical with existing metrics. To do so, they consider idealised cases as well as actual near-term (up to ten days) deterministic forecasts of the CMEMS ARC-MFC product. The authors arrive at a set of metrics they recommend for future evaluations of ice-edge forecast accuracy.

The paper is a useful contribution to the ongoing discussion of how to verify sea-ice forecasts and fits the scope of Ocean Science. I have quite a number of remarks, most of which are however probably straight-forward to address. There are also numerous grammar errors, many of which are listed under "Technical corrections" below. Overall, I recommend publication of the manuscript in Ocean Science subject to minor

revisions.

* Specific comments

P3L29 and elsewhere: often "grid(s)" is used when "grid cell(s)" is meant. Also, sometimes "nodes" is used instead. I recommend to use "grid cell(s)" consistently (where that is meant, of course). This also holds for the Supplement.

P4Eq7: I suggest to make it explicit that d_o and d_m are not single scalars but sets, if I am not mistaken, by writing the right-hand-side as "max(max(d_o),max(d_m))".

P5Eq8: It seems that statements like "aˆ+ = 0 elsewhere" and "aˆ- = 0 elsewhere" are missing in the upper and lower equation, respectively.

P7Eq17: I am somewhat irritated by this equation. For example, when I substitute (i_k)ˆn (bottom left) into the upper equation, the first term in the brackets becomes 1+k(n+1), which doesn't seem to make much sense. Isn't (i_k)ˆn supposed to stay the same when the sums are evaluated, that is, should the indices be different?

P8L3-9: It might be OK not to repeat the algorithm for the FSS displacement, but at least a qualitative description of how that quantity is derived from the FSS should be provided.

P9L16-17: "the resulting displacement metrics are also reduced substantially from the Reference case to the Modified case, due to the added ice area's proximity to land."; Is this sentence really saying what it's supposed to say? After all, they are still increasing, only much less.

P10L4-7: It seems worth mentioning that the Hausdorff-type metrics do not require remapping, although it seems OK to do it in this study to ensure consistency. This could also be mentioned in the discussion part

P10L23-32: Here I was surprised that the relation between (DˆIE)hat and DˆIE is not mentioned, and also not the relation between (DˆIIEE)hat and DˆIIEE. Likewise, it's

worth to highlight already that DˆIIEE and (DˆIE)hat are very similar. You elaborate on this only in the next section, and I think this is an interesting outcome that gives confidence about the robustness of these two metrics which are technically derived in quite different ways.

P11L22-26: What can be concluded from the comparison of the two different observational products? Can this help to understand the relatively large errors that are present already in the initial states? It would be good to comment on this.

P12L8: "This was to be expected"; Actually, I would not have expected such a close match, given the considerably different approach to derive these two metrics.

P12L18: "50 such pairs" -> "50 out of 105 pairs" (correct?)

P14L5-6: Regarding the maps, these would be examples of past performance rather some kinds of averages, which I wouldn't know how that should work, right? Or maps showing the errors for the latest previous forecasts (making use of the slow decorrelation)?

P14L14-18: I have difficulties to understand this paragraph on the usefulness of providing FSS in addition. I suggest to either explain a bit more, or to remove this paragrapgh.

P15L1-3: Is the Palerme et al. paper published now? It's not ideal to base an important final recommendation partly on a not-yet-published paper.

P15L3-4: "We have shown that the deterioration in the forecast quality is moderate for these lead times"; Again, I think there should be some discussion on why there is such a relatively large initial error (which is partly responsible for this slow initial error growth, I would say).

Figure2: Is A- and A+ the wrong way around here? Shouldn't A+ be the part where the model/forecast has too much ice?

Figure5: A statement on the units of the y-axis is missing.

Supplement EqS2-S4: It appears strange to me to use the areas (aˆia) as weights when averaging over the different segments the edge consists of. Wouldn't it make more sense to use the lengths l as weights? In case of S3, and neglecting A_0, this would yield simply D... = sum(a)/sum(l) . Also, for the same reason, the term A_0 seems a bit arbitrary: this one would converge to zero for increasing resolution, right? I am also suspecting that this awkward weighting is the reason why the hat-versions of DˆIIEE are by such a large factor larger than those without hat.

* Technical corrections

P1L19-20: "for appropriate" -> "for an appropriate"

P2L4: "distance of the southern route" -> "length of the southern route" (or other way to correct this)

P2L10: "sensitivity test for scale" -> "sensitivity tests for scale"

P2L16: "Carriers" -> "Carrieres"

P2L20: "system" -> "systems"

P2L21-23: I would argue that the scale-dependence is not the only reason for using the FSS; rather, it's the idea of fuzzy verification, acknowledging that the time and location of certain features can't be forecast exactly but rather in a statistical sense.

P3L10: "situations which leads" -> "situations which lead"

P3L12: "results ... is given" -> "results ... are given"

P3L22: "constitutes" -> "constitute"

P3L28: "cells M" -> "cells in product M"

P5L16: "instroduces" -> "introduce(d)"

P5L20: "provides" -> "provide"

[Figure]

P9L12: "introduces" -> "introduced"

P10L1: "Thurday" -> "Thursday"

P11L2: "have been" -> "has been"

P12L2: "is criteron" -> add article

P13L9: "nominator" -> "numerator"

P13L23: "in minds" -> "in mind"

P14L22: "in Supplementary Information document" -> "in the Supplementary Information document"

P15L1: "uo to" -> "up to"

Figure6 (caption): "displaced" -> "displayed"
* * *

---

## Referee Comment (RC2) · Anonymous Referee #2 · 12 Mar 2019

1. General comments

This article proposes to review and test existing published metrics for sea ice edge forecast verification. I would have mentioned in the title that this is a review or evaluation of a series of existing metrics. After an introduction that presents the state-of-the-art published results on sea ice edge validation, the section 2 provides a description of the metrics that are tested in this article. Then, evaluation and relative behaviour of these metrics is performed using basic synthetic cases (section3), then 2 five day forecast synoptic cases in 2017 in the Arctic Ocean (section 4), then the full series of weekly five-day forecast over the full year 2017 (section 5). Discussion and recommendations are presented in section 6.

A supplementary information is provided that give additional description of the metrics, together with technical details.

[Figure]

This is a technical science-based article that should be published. The topic is of high relevance for the sea ice forecast community. However this article is poorly written and organised in section 2, and should deserve a revision to better introduce every metrics in this section, merging with some part of the supplementary paper material, then have a proper annex for very technical aspects. The metrics evaluation and discussion are also lacking an assessment of the robustness of the metrics.

2. Specific comments

2.1 Scientific significance The author are selecting the existing state-of-the-art metrics used by the sea ice forecast community. The overall metrics evaluation gives some guidance for any forecasting centre to perform similar tests and define adequate metrics. And the recommended metrics are based on well justified discussions.

2.2 Scientific quality The authors are choosing three kind of test-bed experiment. One "theoretical" and synthetic. Then two synoptic cases on the Arctic Ocean, at two different and relevant periods, with different sea ice distributions. Then one year (2017) of statistics established every week. All metrics are tested the same way, and the observation reference data set are ice charts, not used by the forecasting system assimilation mechanism. Only robustness of each metrics should be addressed in addition, either referring to existing published work, or discussed in this article. For instance, the 2017 statistics could be used to test the robustness of the metrics through MonteCarlo or boot-strapping approaches.

2.3 Presentation quality Section 2 needs to be extensively revised. The authors refer to some existing articles for each metrics presented: they should take example of these articles in the way metrics are scientifically, then technically presented. Because sea ice edge and sea ice edge displacement are complicated geometric features that need to be correctly explained. For instance, the authors refer to Roberts and Lean (2008) for the FSS definition. For instance this article says "The purpose of this verification method is to obtain a measure of how forecast skill varies with spatial scale in a

way that can be intuitively understood by users and is also directly applicable for post-processing". We would appreciate some similar explanation in section 2. Hence, the Roberts and Lean (2008) article is very easy to read and understand, because there is a substantial effort to describe the metrics in their physical meaning, give some figures or schematics to explain the geometry, and then provide some mathematical definition: the ration of text versus equation is maybe 10 sentences for a given equation. In the present articles's section 2, there is approximately one equation after 1-2 sentences. The reader gets incomplete understanding of the essence of each metrics, in particular if the reader is not used to formal mathematical definitions (all geoscientists don't have strong mathematical background). However, there is this effort, with the supplementary document to provide more technical explanation. I strongly recommend to merge this document inside the section 2, and if needed, to separate some very technical aspect in an annex at the end of the article.

2.4 Overall evaluation:

1. Does the paper address relevant scientific questions within the scope of OS? YES

2. Does the paper present novel concepts, ideas, tools, or data? YES in the sense that if recommend some adequate metrics, among all existing metrics, for sea-ice edge forecast verification

3. Are substantial conclusions reached? YES the recommendation are clear and straightforward

4. Are the scientific methods and assumptions valid and clearly outlined? YES the evaluation of the different kind of metrics is based on different case studies, and results are scientifically justified

5. Are the results sufficient to support the interpretations and conclusions? YES conclusions of the authors and proposition of the selected metrics are robust

6 is the description of experiments and calculations sufficiently complete and precise

to allow their reproduction by fellow scientists (traceability of results)? Partly, due to the fact that section 2 should provide more comprehensive description of each metrics, in order to facilite their reproduction. . . unless reader directly refer to the reference article.

7. Do the authors give proper credit to related work and clearly indicate their own new/original contribution? YES

8. Does the title clearly reflect the contents of the paper? NO the title should mention that this article provides an evaluation of several metrics, and eventually give some recommendations for sea ice edge forecast verification

9. Does the abstract provide a concise and complete summary? The abstract should be slightly improve to really describe the article content

10. Is the overall presentation well structured and clear? NO section 2 needs to be revisited

11. Is the language fluent and precise? NO, there are many shortcuts and some statements are difficult to understand, even if english is correct

12. Are mathematical formulae, symbols, abbreviations, and units correctly defined and used? NO as mentioned above. Section 2 needs to be revisited

13. Should any parts of the paper (text, formulae, figures, tables) be clarified, reduced, combined, or eliminated? YES section 2, merged with the supplementary Information article

14. Are the number and quality of references appropriate? YES

15. Is the amount and quality of supplementary material appropriate? NO needs to be re-organized, with maybe a technical annex

3. Technical and detailed review:

3.1 Abstract: Line 7: Sentence not clear, in particular with the confusing use of "concentrated": "Such information is traditionally available as a set of metrics that provide a concentrated assessment of the information quality."

L14: "These metrics are analyzed in synthetic examples, in selected cases of actual forecasts, and for a full year of weekly forecast bulletins" This sentence is also confusing: are analyses performed separately for 1) synthetic examples ; 2) few real cases ; 3) a full year of weekly forecast ? Or only one kind of analyses on selected forecast among 1 year of weekly bulletins in some synthetic cases ?

3.2 Article:

L8 p2: Is Melsom et al. (2011) reference easily available ?

L9 p2: The reference Palerme et al (2019) is only submitted: not available for readers at this stage

L15 p2: In these two sentences, are you mentioning statistics of the sea ice extent per se, or statistics of erroneous determination by forecasting centres of the sea ice extent quantity ? This is confusing, also the introduction of "contingency table" made need some more detailed explanation for non-expert reader.

L16 p2: Carriers et al., 2017 reference: TYPO, this is Tom Carrieres, as mentioned in the reference list page 16... fund in https://www.cambridge.org/core/books/sea-ice-analysis-and-forecasting/B74BD33160B03EE1FA77CC9BB80E7DA7

L16 p2: "integral quantities" of what, please clarify.

L24 p2: Not sure that the CMEMS, funded by European Commission DG Grow as part of the Copernicus Program can be defined as a "pan-European project".

L26 p2: CMEMS forecast modelling tools are not limited to "circulation models" : biogeochemical models, wave models. . ..

L28-30 p2: number of production centres: please update following what is presented at http://marine.copernicus.eu/about-us/about-producers/

L5-7 p3: "As we demonstrate in this study, the assessment of quality of the forecasted ice edge position is highly sensitive to the definition of metrics, and to some degree uncertainty due to differences in observational products. The amount of available data is not a limiting factor in this context" This sentence is a concluding statement that should not appear this way in the introduction of this article.

L16 p3: Please rephrase. You mean "between" model and observed quantities. And "eg" looks not adequate here: this is not an example among many. . . It is your purpose to investigate discrepancies between Model and Observed estimates of sea ice edge position.

L18 p3: "grid properties". . . you mean here "grid characteristics" ? "properties" might be more general

L25 p3: In equation (1) please define the "logical AND" symbol that might not be known by all readers

L27 p3: "We also introduce the metric position of grid cell" confusing. Do you refer to the geographical coordinates in a given frame of the cell i,j ?

L29 p3: "Next, for each edge grid cell in each product, we find the distance to the nearest edge grid in the alternative product." Again confusing. Why not saying . . . for each grid cell in the model product, we find the distance in the observed product, or vice-versa ? You have just defined above O and M, and it is not clear to what refers "alternative"

L30 p3: Why introducing "Ealt" when just above you have introduced "Eo" ?

L1 p4: still confusing: what to call the "reference product" ? M or O ?

L2 p4: Equation (2) looks like the Euclidian distance between a given ice edge position between the "alt" product (not clear as mentioned above) and the "reference product" (also not clear) QUESTION: how are associated the ice edge cells between the 2 compared products ? I assume that for a given cell in the first products, several cells

could corresponds in the second product.

L3-5 p4: Not clear if separating situation with/without considering ocean/land boundaries need to be discussed by providing equation (3), similar to equation (2). Maybe just including the ocean/land node point when presenting the detailed explanation on the way this metrics is computed might be sufficient ?

L9 p4: Here the confusion mentioned above clearly appears: Equation 4,5,6 contain reference to "M" and "O" while reader can believe that "Ealt" was "O" .

L23-24 p4: Again, not sure this is useful.

L10-15 p4: The two metrics should be discussed: in practice what do they inform on ? In particular "A+ - A-"

L21 p5: Already mentioned above: the author is Tom Carrieres, not "Carriers"

L1-6 p6: For the sake of simplicity, some diagrams could have been provided, summarising the different configurations of grid cell with/without ice edge and the way the length is determined

L7 p7: "Next, we introduce the coarse grid ice edge fraction for a neighbourhood with an extent of n grid cells as" This definition deserve much more explanation, because this is key-definition to understand equations 17 to 20. "with an extent of n grid cells" is not clear to me, and I imagine for many readers, unless reading the Roberts and Lean (2008), what I have done the shortcut of the present text. Please, give more comprehensive definition before your equations.

L1 p8: It is unfortunate that the supplement explanations are not directly introduced in the article: this is the way Roberts and Lean (2008) proceeded to give shape of their explanation and equations. This should be done in the present article.

L25 p8: "We will demonstrate in Sect.s 4 and 5 below that differences which are qualitatively similar to the Modified case are important to leading order for the quality assessment of the ice edge position in the forecasts from CMEMS ARC MFC". typo in "ÂăSect.sÂă" Again the authors introduce here, too shortly, some conclusions obtained later on in this article. This is rather difficult to follow and confusing.

L29 p8: "and the main purpose of this document is to present metrics for the separation in this set of lines" Again very difficult to understand. Document ? This particular example of Fig 1 ? The full article ? lines. . . the ice edge lines ? a line of discussion ?

L6 p9: "From experience, we know that discrepancies where sea ice emerges or disappears at a distance from other ice covered regions arise from time to time" Not clear. Please explain and/or re-phrase

L10 p9: "Since an additional discrepancy between the observations and model results has been introduced at a large distance, this change is according to our expectations". Not clear. Please explain and/or re-phrase

L23 p9: the CMEMS acronym is already provided.

L6 p10: Typo: overlaid.

L8 p10: "In order to explore how sea ice edge metrics from actual forecasts and observations are affected by changing conditions" .. Not clear to what refers "conditions". Please explain and/or re-phrase.

L2 p11: Figure 4 horizontal axes: problem with the time labels on my PDF version. And labels (a) and (b) do not appear in my PDF version.

L6-7 p11: "which reveal that the sea ice extent is larger in the ice chart product than in the model product." Also mentioning that this brings the negative values of fig 4b.

L14 p11: I recommend to include section S1.1 into the main article.

L23-25: these statistics of comparison between ice concentration assimilated product and ice charts should be added to Table 3,4, wherever they can appear. . . This would be more readable.

L29 p11: Figure 5: In my PDF version, label (a) and (b) are mission in the figures, and it should be more readable to add x- and y-axis label titles. . .. Also some x-axis label numbers are missing (only 1, 2, 5). What happens in both figures for lead-time days 2 to 5 ? Why curves are dashed lines and x-ticks missing (in may PDF version)?

L29-30 p11: "We also note that results for the two metrics in group 2 nearly overlap at all lead times" referring here to curves blue and red would be more readable .

L1-4 p12: "The FSS scores reveal that useful forecasts with a five day lead time are obtained at a scale of about 90x90 km, when the FSS reaches a value of 0.5 (which is criterion recommended by Skok and Roberts (2016)). When comparing with the microwave data, the FSS is well above 0.5 for a neighbourhood extent n = 5 (not shown), corresponding to useful data at a scale of approximately 60x60 km." Here It would have been interesting, with the 2017 comparison, to show the asymptotic behaviour of FSS discussed in Roberts and Lean (2008). It is also interesting to notice the higher resolution quality of the ice concentration (60km useful scales) compared to model results (90km useful scales).

L16 p12: "by systematically computing the correlation coefficients between all possible sets of two displacement metrics" This definition is not clear. Here some more explanation of equation would be useful.

L20-22 p12: Not clear to what these four group refers. . . high, low correlation between them ? Please explain.

L9 p14: this is the first time robustness of the metrics is discussed. As mentioned in the general review comments, there is a lack in this article of robustness assessment of the different metrics (eg, using bootstrap methodology over the 2017 data set).

L20 p14: Sea Ice metrics computed on specific areas was already presented in the GODAE validation article: Hernandez, F., and Coauthors, 2009: Validation and intercomparison studies within GODAE. Oceanography Magazine, 22, 128-143.

http://dx.doi.org/10.5670/oceanog.2009.71

3.3 Supplementary information review:

L6-10 p2: Here a diagram/figure showing the 2 rectangles, and their overlapping area

––––––––––––––––––––––––––––––

---

## Author Response (AR1)

**Authors' response to the OS Editorial Board**

Dear Sir, Madame

We have now completed the revision of our manuscript Validation metrics for ice edge position forecasts which we hereby submit in its final form to Copernicus Publications - Ocean Science; Special Issue: "The Copernicus Marine Environment Monitoring Service (CMEMS): scientific advances".

We have completed revising our original submission, based on comments and suggestions by the referees. Our detailed and itemized responses to both referee statements follow, starting on the next page. Their efforts helped us revise our original submission in a way that we find was highly beneficiary to the quality of our work. We are profoundly grateful for the referees' efforts. Note that a modest additional reorganization and editing was performed on our own initiative, notably moving a section from P11L3-10 to P12L3-10 (referring to the mark-up document).

On behalf of all authors, Arne Melsom

**Authors' response to Referee Comment 1**

We would like to thank the referee for taking the time to carefully read our manuscript and provide a large number of suggestions and comments which we find very useful for the present revision of our manuscript.

Please find our detailed responses to all specific comments below, and note that while we have followed the referee's advise on most of the items, there are a few upon which we have not acted. Initial page and line numbers below (in bold) and comments (in italics) are repeated from the referee's document. This is followed by our response (in regular font) and, when relevant, reference to where changes can be found in the mark-up version of the revised submission (in italic bold).

**P3L29 and elsewhere**

often "grid(s)" is used when "grid cell(s)" is meant. Also, some- times "nodes" is used instead. I recommend to use "grid cell(s)" consistently (where that is meant, of course). This also holds for the Supplement.

Where applicable we have replaced "node(s)" and "grid(s)" with "grid cell(s)" (also in the Supplementary Information document). An example where "grid" was not modified is when referring to a "stereographic grid".

**P4Eq7**

I suggest to make it explicit that  $d_o$  and  $d_m$  are not single scalars but sets, if I am not mistaken, by writing the right-hand-side as "max $(max(d_o), max(d_m))$ ".

Even though the contents of Eq. 7 is not affected, we have elected to follow the referee's suggestion and modified the equation as recommended. *P5L16*

**P5Eq8**

It seems that statements like " $a^+ = 0$  elsewhere" and " $a^- = 0$  elsewhere" are missing in the upper and lower equation, respectively.

The referee is correct, and Eq. 8 has been rewritten accordingly. P5L25-P6L1

**P7Eq17**

I am somewhat irritated by this equation. For example, when I substitute  $i_k^n$  (bottom left) into the upper equation, the first term in the brackets becomes  $1 + k \cdot (n + 1)$ , which doesn't seem to make much sense. Isnt  $i_k^n$  supposed to stay the same when the sums are evaluated, that is, should the indices be different?

The referee's irritation regarding Eq. 17 is highly justified. We have taken two actions related to this issue. First, in our original manuscript *I* as defined by Eq. 16 is a grid cell quantity. Since all other quantities for the grid cell level are in lower case, this was unfortunate. Hence, we have replaced *I* by  $\lambda$  in the present revision. Second, the reviewer rightly rejects the use of *e.g.*  $i_k^n$  in Eq. 17, the correct here is  $i^n$ . The equation has been corrected accordingly. *P8L25-26*

**P8L3-9**

It might be OK not to repeat the algorithm for the FSS displacement, but at least a qualitative description of how that quantity is derived from the FSS should be provided.

We have rewritten Sect. 2.3 to provide more general information on the FSS metric in the first paragraphs, and we also provide an approximate expression for the relation between FSS values and FSS displacement lengths in the final paragraph. *P8L7-13, P10L15-19*

**P9L16-17**

"the resulting displacement metrics are also reduced substantially from the Reference case to the Modified case, due to the added ice areas proximity to land."; Is this sentence really saying what its supposed to say? After all, they are still increasing, only much less.

The referee is correct, and the sentence in question have been rewritten accordingly. P11L31-32

**P10L4-7**

It seems worth mentioning that the Hausdorff-type metrics do not require remapping, although it seems OK to do it in this study to ensure consistency. This could also be mentioned in the discussion part

The referee is correct that Hausdorff-type metrics do not require remapping. The main contrasts between our approach and that of some other investigations is that we treat the ice edge as being composed of grid cells, rather than one-dimensional curves. We have added a paragraph on this topic (the second paragraph in Sect. 2). Moreover, while it is possible to define displacement metrics also for sets of grid cells given on different resolutions and projections, there are then complications related to representativeness that we find to be somewhat beyond the scope of the present study. *P3L29-31*

**P10L23-32**

Here I was surprised that the relation between  $\widehat{D^{IE}}$  and  $D^{IE}$  is not mentioned, and also not the relation between  $\widehat{D^{IEE}}$  and  $D^{IIEE}$ . Likewise, its worth to highlight already that  $D^{IIEE}$  and  $\widehat{D^{IE}}$  are very similar. You elaborate on this only in the next section, and I think this is an interesting outcome that gives confidence about the robustness of these two metrics which are technically derived in quite different ways.

As suggested here by the referee we have added a section (second to last in Sect. 4) where results for various metrics from the two forecasts are mentioned. *P13L19-24*

**P11L22-26**

What can be concluded from the comparison of the two different observational products? Can this help to understand the relatively large errors that are present already in the initial states? It would be good to comment on this.

The referee is correct about the impact of the contrasts between the assimilated microwave data and the ice chart data used for validation. We have added a sentence about this in the paragraph in question, and also in the following section. However, we refer to initial differences as 'deviations' rather than 'errors' since the two observational products in question have their separate strengths and weaknesses, so the 'truth' is not known. Finally, additional results from the comparison between the two observational products are now given in the Supplementary Information. *P14L18-21,P14L29-P15L2, Sect. S2, Tables S1,S2, Fig. S3*

**P12L8**

"This was to be expected"; Actually, I would not have expected such a close match, given the considerably different approach to derive these two metrics.

We admit that the expected relationship between displacement metrics should have been explained more carefully. In the present revision we have included a discussion of idealized cases in the beginning of Sect. 6 which should shed light on this topic. *P15L9-15*

**P12L18**

"50 such pairs" -> "50 out of 105 pairs" (correct?)

Yes, it's 50 out of 105 pairs. This is stated explicitly in the revised manuscript. P16L3-4

**P14L5-6**

Regarding the maps, these would be examples of past performance rather some kinds of averages, which I wouldnt know how that should work, right? Or maps showing the errors for the latest previous forecasts (making use of the slow decorrelation)?

Our recommendation is due to the latter, *i.e.* the long decorrelation time scale. In order to explain this better, we have rearranged Sect. 6.3 and rewritten the sentence in question. *P17L21-23, P18L12-15*

**P14L14-18**

I have difficulties to understand this paragraph on the usefulness of providing FSS in addition. I suggest to either explain a bit more, or to remove this paragraph.

The sentence concerning steepness of 0.5-crossing was not documented, and may thus be incorrect.

This sentence has been removed. **P17L32-P18L2** However, the application of FSS for examination of systems with different resolutions is at the core of this metric, and has been described thoroughly in papers that we cite, see *e.g.* Roberts and Lean. This is also stated in the Introduction section of the present manuscript. Based on suggestions from another referee the presentation of the FSS metric has been re-arranged in this revision.

**P15L1-3**

Is the Palerme et al. paper published now? Its not ideal to base an important final recommendation partly on a not-yet-published paper.

Palerme et al. is not yet published, but a revised manuscript based on a 'minor revision' recommendation has been submitted. However, we disagree that our recommendation is partly based on this study. Palerme et al. was mentioned here for context. Nevertheless, we have moved this sentence to the Introduction section, where it fits nicely in a paragraph where relevant literature is listed. *P2L20-21, P18L27-29*

**P15L3-4**

"We have shown that the deterioration in the forecast quality is moderate for these lead times"; Again, I think there should be some discussion on why there is such a relatively large initial error (which is partly responsible for this slow initial error growth, I would say).

A discussion on the impact of initial errors, or rather deviations, is provided in Sect. 5 in the revised manuscript, see our reply to item **P11L22-26** above.

**Figure2**

Is  $A^-$  and  $A^+$  the wrong way around here? Shouldnt  $A^+$  be the part where the model/forecast has too much ice?

The referee asks if there is an error in the color shading in Fig. 2 in our original submission, and we have indeed made the mistake that the referee has spotted. We are very grateful that the referee pointed us to our mistake. In the revised manuscript the error has been corrected. We can add that we double-checked Fig. 5 (Fig. 3 in the original submission), and found that this did *not* contain the same mistake. *Fig. 4*

**Figure5**

A statement on the units of the y-axis is missing.

The units of the y-axis is now given in the caption. *Fig.* 7

**EqS2-S4**

It appears strange to me to use the areas  $(a^{ia})$  as weights when averaging over the different segments the edge consists of. Wouldnt it make more sense to use the lengths I as weights? In case of S3, and neglecting  $A_0$ , this would yield simply  $D_{...} = \sum a / \sum l$ . Also, for the same reason, the term  $A_0$  seems a bit arbitrary: this one would converge to zero for increasing resolution, right? I am also suspecting that this awkward weighting is the reason why the hat-versions of  $D^{IIEE}$  are by such a large factor larger than those without hat.

The application of area weights was introduced in order to highlight effects of the geometry of IIEE areas, as stated in Sect. 2.2.2. With the referee's suggestion (e.g.  $\sum a^{ia} / \sum l^{ia}$ ) the metrics would essentially give the same information as the original  $D^{IIEE}$  metrics: consider the fraction  $\widehat{D_{AVG}^{IIEE}} / D_{AVG}^{IIEE}$  in the three idealized cases we present. For  $\nu = 1/4$  the resulting fractions are 1.5, 1.7 and 1.35, respectively. Adopting the referee's suggestion we find the set of corresponding values to be 1.38, 1.36 and 1.36. For  $\nu = 4$  the resulting fractions are 3, 2.5 and 2.3, while the referee's alternative yields fractions of 1.17, 1.13, 1.13. Moreover, the term  $A_0$  is not arbitrary: in the case of two identical ice edges, dropping  $A_0$  will lead to an ill-defined value for  $\widehat{D_{AVG}^{IIEE}}$  since with no  $A_0$  it becomes 0/0. Note that  $A_0$  is the area of all grid cells where the products overlap, the sentence in question has been rewritten to make this clear. *S-P1L20*

**Technical corrections**

All of these items have been corrected according to the referee's advice.

**Authors' response to Referee Comment 2**

We are grateful for the referee's careful consideration of our manuscript and provision of a large number of comments which we find very helpful for the present revision of our manuscript, particularly for Sect. 2.

The referee has a number of suggestion of expanding the main article, *e.g.* (i) by moving material from the Supplementary Information to Sect. 2, (ii) by assessing robustness of metrics through MonteCarlo or boot-strapping approaches in Sect. 5, and (iii) by including more results from a comparison between the microwave product and ice charts in Sect. 5. Our general response is that expansions of the main article should not include material that for a large part becomes distractions from the topic, which is an evaluation of metrics for sea ice edge position forecasts. Following this guideline, we have chosen to comply with the referee's advice concerning (i) and (ii). Item (iii) is also addressed, but additional results are given in the Supplementary Information.

We are advised to change the title so that it includes references to evaluation of several metrics, and subsequent provision of recommendations for sea ice edge forecast verification. We disagree. The title should not be a long sentence, but provide enough information that the attention of an interested reader would be caught from a contents listing or from a web search lookup. We believe that our title serves this purpose. The fact that evaluations and recommendations are given follows implicitely from the title as is. The abstract has been rewritten slightly, following the relevant detailed comments made by the referee.

Please find our detailed responses to all specific comments below, and note that while we have followed the referee's advise on most of the items, there are a few items upon which we have not acted. Initial page and line numbers below (in bold) and comments (in italics) are repeated from the referee's document. This is followed by our response (in regular font) and, when relevant, reference to where changes can be found in the mark-up version of the revised submission (in italic bold).

**P1L7**

Sentence not clear, in particular with the confusing use of 'concentrated': "Such information is traditionally available as a set of metrics that provide a concentrated assessment of the information quality."

Here, 'concentrated' referred to the fact that a metric is a single number that provides a condensed assessment of a two-dimensional field. Since this is basically the nature of a metric, we have taken the referee's advise an removed this word in the revised document. *P1L8*

**P1L14**

"These metrics are analyzed in synthetic examples, in selected cases of actual forecasts, and for a full year of weekly forecast bulletins" This sentence is also confusing: are analyses performed separately for 1) synthetic examples ; 2) few real cases; 3) a full year of weekly forecast? Or only one kind of analyses on selected forecast among 1 year of weekly bulletins in some synthetic cases?

The sentence in question has been rephrased to make its content more clear. P1L15

**P2L8**

Is Melsom et al. (2011) reference easily available?

We have added a web reference from which Melsom et al. (2011) is available. Furthermore, we note that Melsom et al. (2011) was cited by one of the references in the present study (Goessling et al., 2016, GRL). *P20L29*

**P2L9**

**The reference Palerme et al (2019) is only submitted: not available for readers at this stage**

Referee statements to the submission to GRL of Palerme et al. have been provided. The editor concludes that the manuscript "may be suitable for publication after minor revisions". A revised manuscript was submitted to GRL earlier this week. We have not been able to find a policy statement regarding when a citation is acceptable, so we leave it as is until we are advised differently by the editor or the technical editor.

**P2L15**

In these two sentences, are you mentioning statistics of the sea ice extent per se, or statistics of erroneous determination by forecasting centres of the sea ice extent quantity? This is confusing, also the introduction of 'contingency table' made need some more detailed explanation for non-expert reader.

Model *vs.* observation contingency tables provides results for the sea ice extent for each of the two product, as well as for the sea ice extent mismatch between the product. However, details regarding contingency tables are not appropriate in the Introduction section. Accordingly, we have added some sentences to explain this matter in Sect. 2.2. *P6L19-24*

**P2L16 + P5L21**

*Carriers* et al., 2017 reference: TYPO, this is Tom Carrieres, as mentioned in the reference list page 16... found in

https://www.cambridge.org/core/books/sea-ice-analysis-and-forecasting/B74BD33160B03EE1FA77CC9BB80E7DA7 + Already mentioned above: the author is Tom Carrieres, not 'Carriers'

The reference has been corrected. The duplicity was due to using the LaTex citation feature. Note that these changes are not highlighted in the mark-up document since they resulted from latex citation code rather than an explicit typographical mistake. *P2L17, P6L19*

**P2L16**

'integral quantities' of what, please clarify.

The integral quantity here is the sea ice extent. We have rewritten the sentence to make this clear. *P2L17-18*

**P2L24**

Not sure that the CMEMS, funded by European Commission DG Grow as part of the Copernicus Program can be defined as a 'pan-European project'.

We have substituted 'pan-European project' with the description given by the EU Copernicus Programme. *P2L29-30*

**P2L26**

CMEMS forecast modelling tools are not limited to "circulation models" : biogeochemical models, wave models...

We have added other model systems to the list, as suggested by the referee. *P2L31-32*

**P2L28-30**

number of production centres: please update following what is presented at http://marine.copernicus.eu/aboutus/about-producers/

The number of CMEMS centers listed in the text has been updated. P3L1

**P3L5-7**

"As we demonstrate in this study, the assessment of quality of the forecasted ice edge position is highly sensitive to the definition of metrics, and to some degree uncertainty due to differences in observational products. The amount of available data is not a limiting factor in this context" This sentence is a concluding statement that should not appear this way in the introduction of this article. We have rewritten these sentences along the lines suggested by the referee. **P3L12-15**

**P3L16**

Please rephrase. You mean 'between' model and observed quantities. And 'eg' looks not adequate here: this is not an example among many... It is your purpose to investigate discrepancies between Model and Observed estimates of sea ice edge position.

We have replaced 'in' by 'between' (P3L24). Further, the referee implies that our analysis is limited

to comparisons between model results on one hand and observations on the other. This is incorrect. Metrics like the ones we examine are also used when comparing results for ice edge position between different observational products, which is what we do in Sects. 5 and S2 where we compare ice charts with a microwave product.

**P3L18**

*'grid properties'... you mean here 'grid characteristics'? 'properties' might be more general* We have rewritten 'grid properties' as 'grid cell quantities'. *P3L26-27*

**P3L25**

In equation (1) please define the 'logical AND' symbol that might not be known by all readers

A statement on the symbol  $\land$  used for logical AND has been added after Eq. (1). *P4L6*

**P3L27**

"We also introduce the metric position of grid cell" confusing. Do you refer to the geographical coordinates in a given frame of the cell i,j?

We have rewritten 'metric position' as 'coordinate position'. This is not the geographical longitude, latitude position, but the coordinate position from origo in a projection plane. *P4L7-8,15-16*

**P3L29 + P3L30 + P4L1 + P4L9**

"Next, for each edge grid cell in each product, we find the distance to the nearest edge grid in the alternative product." Again confusing. Why not saying ... for each grid cell in the model product, we find the distance in the observed product, or vice-versa? You have just defined above O and M, and it is not clear to what refers 'alternative'

+ Why introducing 'Ealt' when just above you have introduced 'Eo'?

+ still confusing: what to call the 'reference product'? M or O?

+ Here the confusion mentioned above clearly appears: Equation 4,5,6 contain reference to 'M' and 'O' while reader can believe that 'Ealt' was 'O'.

Following these suggestions and comments, we have removed references to the 'alternative product' and 'reference product' and rewritten Sect. 2.1 accordingly. *P4L10-27*

**P4L2**

Equation (2) looks like the Euclidian distance between a given ice edge position between the 'alt' product (not clear as mentioned above) and the 'reference product' (also not clear) QUESTION: how are associated the ice edge cells between the 2 compared products? I assume that for a given cell in the first products, several cells could corresponds in the second product.

A statement on the symbol  $\forall$  used for the FOR ALL operator has been added after Eq. (2). min is the minimum function (applied to all distances to all grid cells in the second product). **P4L15**

**P4L3-5 + P4L23-24**

Not clear if separating situation with/without considering ocean/land boundaries need to be discussed by providing equation (3), similar to equation (2). Maybe just including the ocean/land node point when presenting the detailed explanation on the way this metrics is computed might be sufficient? + Again, not sure this is useful.

A good number of the referee's comments and suggestions ask for more information, and we think rightly so in most cases. However, here the referee asks us to omit information as removal of Eq. (3) is recommended, and then the way that the resulting metrics are introduced after Eq. (7) is criticized. We find that keeping Eq. (3) is an approach that is more in line with the general level of detail in the manuscript, even more so for the present revision than for the initial submission. By keeping Eq. (3) we find that sufficient information is provided regarding the separation between the metrics defined by Eq. (4)-(7), thus these are not repeated for the hatted metrics counterpart. Hence, no action has been taken in response to these items.

**P5L10-151**

The two metrics should be discussed: in practice what do they inform on? In particular ' $A^+$  -  $A^-$ '

<sup>1Erroneously listed as P4L10-15 in the referee statement

Here,  $A^+$  and  $A^-$  expresses mismatching of the sea ice extent between model and observations. We have added a sentence at the end of the relevant paragraph to make this clear. *P6L7-8*

**P6L1-6**

For the sake of simplicity, some diagrams could have been provided, summarising the different configurations of grid cell with/without ice edge and the way the length is determined

To demonstrate how the ice edge length is determined, we have added a schematic figure and updated the text accordingly. *Fig. 1, P7L6-12*

**P7L7 + P8L1**

"Next, we introduce the coarse grid ice edge fraction for a neighbourhood with an extent of n grid cells as" This definition deserve much more explanation, because this is key-definition to understand equations 17 to 20. "with an extent of n grid cells" is not clear to me, and I imagine for many readers, unless reading the Roberts and Lean (2008), what I have done the shortcut of the present text. Please, give more comprehensive definition before your equations.

+ It is unfortunate that the supplement explanations are not directly introduced in the article: this is the way Roberts and Lean (2008) proceeded to give shape of their explanation and equations. This should be done in the present article.

We include information that was previously provided as Sect. S1.2, now in the main article in Sect. 2.3. This includes a figure (revision of Fig. S2 in the original submission) where the concept of neighbourhood size is exemplified. We believe that this reorganization of text and a figure makes the presentation easier to comprehend for readers who are new to the FSS score. *P8L16-P10L14, Fig. 2*, red text in *S-P3L13-P4L23*

**P8L25**

"We will demonstrate in Sect.s 4 and 5 below that differences which are qualitatively similar to the Modified case are important to leading order for the quality assessment of the ice edge position in the forecasts from CMEMS ARC MFC". typo in '[gibberish]' Again the authors introduce here, too shortly, some conclusions obtained later on in this article. This is rather difficult to follow and confusing.

The sentence in question has been rewritten to point to the subsequent discussion in Sect.s 4 and 5, without stating a conclusion. We cannot find the typo that the referee indicates, likely because the quote on the pdf file with the referee statement appears as gibberish. However, in the event that there is a typo, we are confident that the technical editor will spot it, in the event that our manuscript is accepted for publication. *P11L8-10, P12L8-10*

**P8L29**

"and the main purpose of this document is to present metrics for the separation in this set of lines" Again very difficult to understand. Document ? This particular example of Fig 1? The full article? lines... the ice edge lines? a line of discussion?

The "document" refers to the entire paper. The sentence in question has been rewritten to better reflect our ambition. *P11L12-13*

**P9L6**

"From experience, we know that discrepancies where sea ice emerges or disappears at a distance from other ice covered regions arise from time to time" Not clear. Please explain and/or re-phrase

To make clear which experience we refer to, we have added "in an operational sea ice forecasting service" at the end of the sentence in question. *P11L22*

**P9L10**

"Since an additional discrepancy between the observations and model results has been introduced at a large distance, this change is according to our expectations". Not clear. Please explain and/or re-phrase

The discrepancy we refer to is the one that is described in the paragraph in question, and also in the section in question, as displayed on Fig.s 3 and 4. To make this clear, we have rewritten "Since an additional discrepancy" as "Since the additional discrepancy". *P11L25*

**P9L23**

the CMEMS acronym is already provided.

We have retained the acronym (CMEMS) only in the present revision. P12L12-13

**P10L6**

*Typo: overlaid.* Corrected. **P12L28**

**P10L8**

"In order to explore how sea ice edge metrics from actual forecasts and observations are affected by changing conditions" .. Not clear to what refers 'conditions'. Please explain and/or re-phrase.

Regarding the referee's comment that it is not clear what 'conditions' we refer to, we disagree. The type of conditions we have in mind is stated in the same sentence that the referee only partly cites: "...contrasts of the type that was examined in Sect. 3". No change has been made.

**P11L2**

Figure 4 horizontal axes: problem with the time labels on my PDF version. And labels (a) and (b) do not appear in my PDF version.

There was an error in the compilation of the document that gave rise to the Fig. 4 labeling issues. Fig. 5 had the same problems. We provided corrections in the *Interactive discussion* on 04 Jan 2019, see item 'AC1'.

**P11L6-7**

"which reveal that the sea ice extent is larger in the ice chart product than in the model product." Also mentioning that this brings the negative values of fig 4b.

We now mention the relation to negative values in Fig. 6b. P14L1

**P11L14**

I recommend to include section S1.1 into the main article.

All information that is relevant for the recommended metrics should be explained in the main article. However, our conclusion in Sect. 6.3 is that we don't recommend any of the  $D^{\widehat{IIEE}}$  metrics to be included in operational validation of the sea ice edge position. Hence, on balance our preference is to keep the original organization where details on the  $D^{\widehat{IIEE}}$  metrics' definitions are given in the Supplementary Information document.

**P11L23-25 2**

these statistics of comparison between ice concentration assimilated product and ice charts should be added to Table 3,4, wherever they can appear... This would be more readable.

The purpose of this study is to examine the results for metrics when two products are compared. To keep this focus, we disagree that including results from a third product in tables in the main article. Nevertheless, we wish to provide the reader with some additional results that can shed light on the underlying reasons for discrepancies. So, rather than making any changes in the main article, we add a section (Sect. S2), a figure (Fig. S3), and two tables (Tables S1-S2) in the supplementary information, so that details related to mismatching of the assimilated microwave data and ice charts are available. Blue text in *S-P3L14-P425; Fig. S3; Table S1-S2*

**P11L29**

Figure 5: In my PDF version, label (a) and (b) are mission in the figures, and it should be more readable to add x- and y-axis label titles... Also some x-axis label numbers are missing (only 1, 2, 5). What happens in both figures for lead-time days 2 to 5? Why curves are dashed lines and x-ticks missing (in may PDF version)?

Regarding the labeling issues, we refer to our reply to item **P11L2** above. Dashed lines are used to indicate results that bridge days with no data (ice charts are not produced on Saturdays and Sundays;

<sup>2The page number is missing in the referee's report

see P9L27 in the original submission, P12L16 in the mark-up revision). An explanation has been added in the figure caption. *Fig. 7 caption on P32*

**P11L29-30**

"We also note that results for the two metrics in group 2 nearly overlap at all lead times" referring here to curves blue and red would be more readable.

We now include a reference to the two curves in questions as blue and red, as suggested by the referee. P14L27

**P12L1-4**

"The FSS scores reveal that useful forecasts with a five day lead time are obtained at a scale of about 90x90 km, when the FSS reaches a value of 0.5 (which is criterion recommended by Skok and Roberts (2016)). When comparing with the microwave data, the FSS is well above 0.5 for a neighbourhood extent n = 5 (not shown), corresponding to useful data at a scale of approximately 60x60 km." Here It would have been interesting, with the 2017 comparison, to show the asymptotic behaviour of FSS discussed in Roberts and Lean (2008). It is also interesting to notice the higher resolution quality of the ice concentration (60km useful scales) compared to model results (90km useful scales).

We have moved the comparison between FSS results for the model product and the microwave product to the paragraph where changes as a function of lead time are discussed. (The latexdiff software has split a section in two.) Note that the comparison is now restricted to the period from January to mid-May, which reduces the useful scale. We have also include a figure that displays the FSS score and the asymptote values as defined by Roberts and Lean (2008). *P14L29-P15L5; Fig. 8*

**P12L16**

"by systematically computing the correlation coefficients between all possible sets of two displacement metrics" This definition is not clear. Here some more explanation of equation would be useful.

We have rewritten the sentence, and we have also added some more detail in the text on the next lines. We now refer to this analysis as "systematically computing the correlation coefficients between all possible combinations of displacement metrics time series pairs". *P16L2*

**P12L20-22**

Not clear to what these four group refers... high, low correlation between them ? Please explain.

We now explicitly state which bounds we have used to separate large positive and large negative correlation coefficients from the intermediate and low coefficient values. The absolute values of correlation coefficient meets this criterion for metric pairs inside each of the four groups, as stated in our original submission. *P16L7-8*

**P14L9**

this is the first time robustness of the metrics is discussed. As mentioned in the general review comments, there is a lack in this article of robustness assessment of the different metrics (eg, using bootstrap methodology over the 2017 data set).

We have followed the referee's suggestion, and now include results from bootstrapping in Tables 3 and 4. *Tables 3, 4*

**P14L20**

Sea Ice metrics computed on specific areas was already presented in the GODAE validation article: Hernandez, F., and Coauthors, 2009: Validation and intercomparison studies within GODAE. Oceanography Magazine, 22, 128-143. http://dx.doi.org/10.5670/oceanog.2009.71

We have included the reference to Hernandez et al. (2009) in the present revision. P18L6; P20L17-19

**S-P2L6-10**

Here a diagram/figure showing the 2 rectangles, and their overlapping area

A schematic diagram displaying a sample configuration with rectangular IIEE areas has been included in the present version of the Supplementary Information document. *Fig. S2*

[revised manuscript text omitted]
  $\frac{I_0}{I_0} = \frac{I_1}{\lambda_0}\lambda_n = \lambda_0^1$ . In the example in Fig. 2, a neighbourhood extent of 3 grid cells is indicated by the thick grid lines and for this case, we find

$$\lambda_{O}^{n=3} = \frac{1}{9} \begin{pmatrix} 2 & 1 \\ 0 & 2 \end{pmatrix} \quad ; \quad \lambda_{M}^{n=3} = \frac{1}{9} \begin{pmatrix} 3 & 1 \\ 0 & 3 \end{pmatrix}$$
(18)

The mean square edge fraction error for a neighbourhood extent of n grids becomes grid cells becomes

5
$$MSE^{n} = \frac{1}{N_{x}^{n}N_{y}^{n}} \sum_{i^{n}=1}^{N_{y}^{n}} \sum_{j^{n}=1}^{N_{y}^{n}} \left[ \underline{I}\lambda_{m}^{n}[i^{n},j^{n}] - \underline{I}\lambda_{o}^{n}[i^{n},j^{n}] \right]^{2}$$
 (19)

where  $N_x^n$ ,  $N_y^n$  are the number of the neighbourhood extent  $n \frac{\text{grids}}{\text{grid}} \frac{\text{grid}}{\text{cells}}$  in the x and y directions, respectively. Following Roberts and Lean (2008) we introduce a reference MSE value as the largest possible with the present extent of the edge nodes grid cells

$$MSE_{ref}^{n} = \frac{1}{N_{x}^{n}N_{y}^{n}} \min\left\{ \left[ \sum_{i=1}^{N_{x}^{n}} \sum_{j=1}^{N_{y}^{n}} \lambda_{o}^{n} [i^{n}, j^{n}]^{2} + \sum_{i=1}^{N_{x}^{n}} \sum_{j=1}^{N_{y}^{n}} \lambda_{m}^{n} [i^{n}, j^{n}]^{2} \right], \\ \left[ \sum_{i=1}^{N_{x}^{n}} \sum_{j=1}^{N_{y}^{n}} \left( 1 - \lambda_{o}^{n} [i^{n}, j^{n}] \right)^{2} + \sum_{i=1}^{N_{x}^{n}} \sum_{j=1}^{N_{y}^{n}} \left( 1 - \lambda_{m}^{n} [i^{n}, j^{n}] \right)^{2} \right] \right\}$$
(20)

This expression is a worst case arrangement of hits and misses that takes into account  $e_{e,e}$ , situations where hits outnumber 10 misses. This is a modification of the corresponding definition in Roberts and Lean (2008) whose Eq. 7 allowed for situations with  $MSE_{ref}^{n}$  exceeding 1.

For the skill score with the original  $6 \times 6$  grid in Fig. 2 we have  $MSE^{n=1} = 6/6^2$  and  $MSE^{n=1}_{ref} = 12/6^2$ , while for the n = 3neighbourhood displayed by the thick grid lines we have  $MSE^{n=3} = 2/(2 \cdot 9)^2$  and  $MSE^{n=3}_{ref} = 9/(2 \cdot 9)^2$ .

15 Now, the resolution-dependent fractions skill score is introduced as

$$FSS^{n} = 1 - \frac{MSE^{n}}{MSE^{n}_{ref}}$$
(21)

which has a value of 1 for a perfect forecast for neighbourhood extent  $n (I_m^n = I_o^n \forall i^n, j^n \Rightarrow MSE^n = 0 \lambda_o^n \forall j^n \Rightarrow MSE^n = 0 \lambda_o^n \forall j^n \Rightarrow MSE^n = 0 \lambda_o^n \forall j^n \Rightarrow MSE^n \Rightarrow MS$ and a value of 0 when  $\frac{I_m^n + I_o^n = 0 \quad \forall i^n, j^n \\ \lambda_m^n + \lambda_o^n = 0 \quad \forall i^n, j^n \\ \lambda_m^n + \lambda_o^n = 0 \quad \forall i^n, j^n \\ \lambda_m^n + \lambda_o^n = 0 \quad \forall i^n, j^n \\ \lambda_m^n + \lambda_o^n = 0 \quad \forall i^n, j^n \\ \lambda_m^n + \lambda_o^n = 0 \quad \forall i^n, j^n \\ \lambda_m^n + \lambda_o^n = 0 \quad \forall i^n, j^n \\ \lambda_m^n + \lambda_o^n = 0 \quad \forall i^n, j^n \\ \lambda_m^n + \lambda_o^n = 0 \quad \forall i^n, j^n \\ \lambda_m^n + \lambda_o^n = 0 \quad \forall i^n, j^n \\ \lambda_m^n + \lambda_o^n = 0 \quad \forall i^n, j^n \\ \lambda_m^n + \lambda_o^n = 0 \quad \forall i^n, j^n \\ \lambda_m^n + \lambda_o^n = 0 \quad \forall i^n, j^n \\ \lambda_m^n + \lambda_o^n = 0 \quad \forall i^n, j^n \\ \lambda_m^n + \lambda_o^n = 0 \quad \forall i^n, j^n \\ \lambda_m^n + \lambda_o^n = 0 \quad \forall i^n, j^n \\ \lambda_m^n + \lambda_o^n = 0 \quad \forall i^n, j^n \\ \lambda_m^n + \lambda_o^n = 0 \quad \forall i^n, j^n \\ \lambda_m^n + \lambda_o^n = 0 \quad \forall i^n, j^n \\ \lambda_m^n + \lambda_o^n = 0 \quad \forall i^n, j^n \\ \lambda_m^n + \lambda_o^n = 0 \quad \forall i^n, j^n \\ \lambda_m^n + \lambda_o^n = 0 \quad \forall i^n, j^n \\ \lambda_m^n + \lambda_o^n = 0 \quad \forall i^n, j^n \\ \lambda_m^n + \lambda_o^n = 0 \quad \forall i^n, j^n \\ \lambda_m^n + \lambda_o^n = 0 \quad \forall i^n, j^n \\ \lambda_m^n + \lambda_o^n = 0 \quad \forall i^n, j^n \\ \lambda_m^n + \lambda_o^n = 0 \quad \forall i^n, j^n \\ \lambda_m^n + \lambda_o^n = 0 \quad \forall i^n, j^n \\ \lambda_m^n + \lambda_o^n = 0 \quad \forall i^n, j^n \\ \lambda_m^n + \lambda_o^n = 0 \quad \forall i^n, j^n \\ \lambda_m^n + \lambda_o^n = 0 \quad \forall i^n, j^n \\ \lambda_m^n + \lambda_o^n = 0 \quad \forall i^n, j^n \\ \lambda_m^n + \lambda_o^n = 0 \quad \forall i^n \\ \lambda_m^n + \lambda_o^n = 0 \quad \forall i^n \\ \lambda_m^n + \lambda_o^n = 0 \quad \forall i^n \\ \lambda_m^n + \lambda_o^n = 0 \quad \forall i^n \\ \lambda_m^n + \lambda_o^n = 0 \quad \forall i^n \\ \lambda_m^n + \lambda_o^n = 0 \quad \forall i^n \\ \lambda_m^n + \lambda_o^n = 0 \quad \forall i^n \\ \lambda_m^n + \lambda_o^n = 0 \quad \forall i^n \\ \lambda_m^n + \lambda_o^n = 0 \quad \forall i^n \\ \lambda_m^n + \lambda_o^n = 0 \quad \forall i^n \\ \lambda_m^n + \lambda_m^n = 0 \quad \forall i^n \\ \lambda_m^n + \lambda_m^n = 0 \quad \forall i^n \\ \lambda_m^n + \lambda_m^n = 0 \quad \forall i^n \\ \lambda_m^n + \lambda_m^n = 0 \quad \forall i^n \\ \lambda_m^n + \lambda_m^n = 0 \quad \forall i^n \\ \lambda_m^n + \lambda_m^n = 0 \quad \forall i^n \\ \lambda_m^n + \lambda_m^n = 0 \quad \forall i^n \\ \lambda_m^n + \lambda_m^n = 0 \quad \forall i^n \\ \lambda_m^n + \lambda_m^n = 0 \quad \forall i^n \\ \lambda_m^n + \lambda_m^n = 0 \quad \forall i^n \\ \lambda_m^n + \lambda_m^n = 0 \quad \forall i^n \\ \lambda_m^n + \lambda_m^n = 0 \quad \forall i^n \\ \lambda_m^n + \lambda_m^n = 0 \quad \forall i^n \\ \lambda_m^n + \lambda_m^n = 0 \quad \forall i^n \\ \lambda_m^n + \lambda_m^n = 0 \quad \forall i^n \\ \lambda_m^n + \lambda_m^n = 0 \quad \forall i^n \\ \lambda_m^n + \lambda_m^n = 0 \quad \forall i^n \\ \lambda_m^n + \lambda_m^n = 0 \quad \forall i^n \\ \lambda_m^n + \lambda_m^n = 0 \quad \forall i^n \\ \lambda_m^n + \lambda_m^n = 0 \quad \forall i^n \\ \lambda_m^n + \lambda_m^n = 0 \quad \forall i^n \\ \lambda_m^n + \lambda_m^n = 0 \quad \forall i^n \\ \lambda_m^n + \lambda_m^n = 0 \quad \forall i^n \\ \lambda_m^n = 0 \quad \forall i^n \\ \lambda_m^n + \lambda_m^n = 0 \quad$ definition of  $MSE_{ref}^n$  in Eq. 20 makes the  $
[revised manuscript text omitted]

---

## Author Response (AR2)

**Authors' response to the OS Editorial Board**

Dear Sir, Madam

We have completed the final revision of our manuscript
*Validation metrics for ice edge position forecasts*
to Copernicus Publications - Ocean Science; Special Issue: "The Copernicus Marine Environment Monitoring Service (CMEMS): scientific advances". All files required for the production process are now being uploaded to the File Manager with Copernicus Office user ID 377803.

Regarding the technical matter pointed out in *Report #1* the referee is correct and the citation has been modified accordingly.

There are no specific requests to the authors in *Report #2* but the referee mentions that he/she misses a discussion of the results from the bootstrap tests. We agree that this was a short-coming, and accordingly, we have included a paragraph at the end of the initial text in Section 6 (p. 14, l. 11-18 in the mark-up document that is attached here) where bootstrap analysis results are discussed. In doing so, we realized that the definition of fractions in Tables 3 and 4 was ill posed for bias metrics. This is explained in the new paragraph in Section 6, and the corresponding values have been removed from Tables 3 and 4.

Finally, we have updated the reference to Palerme et al. which was recently accepted for publication in Geophysical Research Letters.

On behalf of all authors,
Arne Melsom

[revised manuscript text omitted]